# SMAN-Bench: A Cross-System Benchmark for Mobile Agents under Single- and Multi-path, Ambiguous, and Noisy Tasks

**Weikai Xu**[14*†], **Zhizheng Jiang**[24*†], **Yuxuan Liu**[34], **Pengzhi Gao**[4], **Wei Liu**[4], **Jian Luan**[4],
**Yuanchun Li**[5], **Yunxin Liu**[5], **Bin Wang**[4], **Bo An**[1‡]
[1]Nanyang Technological University [2]University of Electronic Science and Technology of China
[3]Gaoling School of Artificial Intelligence, Renmin University of China
[4]MiLM Plus, Xiaomi Inc. [5]Institute for AI Industry Research (AIR), Tsinghua University
[*]Equal contribution

## Abstract

VLM-based mobile agents are increasingly popular due to their capabilities to interact with smartphone GUIs and XML-structured texts and complete daily tasks. However, existing online benchmarks fail to provide stable fine-grained reward signals under dynamic environmental changes and neglect the influence of noisy components and interactive instructions. Offline benchmarks evaluate the agents through single-path trajectories, which stand in contrast to the inherently multi-solution characteristics of GUI tasks. To address these limitations, we introduce **SMAN-Bench**, a benchmark designed to evaluate agents under **S**ingle-path, **M**ulti-path, **A**mbiguous, and **N**oisy task settings. We employ a slot-based instruction generation method to match templates with GUI trajectories from an existing, graph-structured, unlabeled mobile corpus. SMAN-Bench includes a common task split, with offline multi-path evaluation to assess the agent's ability to obtain step rewards during task execution. It contains a noisy split based on pop-ups and ad apps, and a contaminated split to simulate a realistic noisy environment. Furthermore, an ambiguous instruction split with preset Q&A interactions is released to evaluate the agent's proactive interaction capabilities. Our evaluation encompasses mobile agent frameworks such as AppAgent-v1, Mobile-Agent-v2, and Mobile-Agent-E, and includes both open-source and closed-source mobile foundation models, as well as several multimodal thinking models. Our code and datasets are available at `https://github.com/gezelligheid0314/SMAN-Bench`.

## 1 Introduction

LLM-based mobile agents (Wang et al., 2023; Ding, 2024) are increasingly popular due to their capability to interact directly with mobile Graphical User Interfaces (GUIs) and their potential to manage daily tasks autonomously. Unfortunately, LLM-based agents struggle to comprehend mobile GUI structures and widget functionality from textual representations such as Visual-Hierarchical, XML, HTML, or Accessible-Visited Trees. Recent studies (Ma et al., 2024; Zhang et al., 2024a) indicate VLMs can provide a more comprehensive understanding of GUIs. This has led to mobile benchmarks replacing foundational models with VLMs, resulting in benchmarks for end-to-end mobile tasks based on GUI pages (Wang et al., 2024c; Xu et al., 2024a; Rawles et al., 2024).

Existing VLM-based Agent benchmarks can be divided into two categories: (1) **Online evaluation** involves the agent executing operations on a real device based on the user's high-level instructions. These benchmarks directly determine the success rate by checking the widget values in the final GUI and allow agents to complete tasks through multiple paths. However, due to the instability of the device environment, such as OS updates, app updates, and user preference records, the step rewards of online benchmarks (Murthy et al., 2024; Deng et al., 2024; Wang et al., 2024c) are fluctuating and

---

[†]Work done during the internship at XiaoMi.

[‡]Bo An is the corresponding author.

Table 1: Comparison of SMAN-Bench to other benchmarks.

| Benchmarks | # Inst. | Language | # Avg Steps | # Screen-shots | Path | Online Environ.? | Ambi. Noise | Interac. Inst. |
|---|---|---|---|---|---|---|---|---|
| PIXELHELP | 187 | EN | 4.2 | ∼800 | Single | ✗ | ✗ | ✗ |
| MOTIF | 480 | EN | 4.5 | ∼21K | Single | ✗ | ✗ | ✗ |
| AMEX | 341 | EN&CN | 12.8 | ∼104K | Single | ✗ | ✗ | ✗ |
| SCREENSPOT | ∼1,200 | EN&CN | 1 | ∼600 | Dot | ✗ | ✗ | ✗ |
| MOBILEAIBENCH | * | EN | * | * | Dot | ✗ | * | ✗ |
| AGENTBENCH | 100 | EN | 20 | ∼2k | Multiple | ✓ | ✗ | ✗ |
| GUI ODYSSEY | 7,735 | EN | 15.4 | * | Single | ✗ | ✗ | ✗ |
| MOBILE-BENCH | 832 | CN | * | 14,144 | Multiple | ✓ | ✗ | ✗ |
| MVISU-BENCH | 404 | EN&CN | * | * | Multiple | ✓ | ✗ | ✓ |
| SPA-BENCH | 340 | EN&CN | * | * | Multiple | ✓ | ✗ | ✗ |
| ANDROIDLAB | 10.5k | EN | 8.98 | 94.3k | Multiple | ✓ | ✗ | ✗ |
| ANDROIDWORLD | 116 | EN | * | * | Multiple | ✓ | ✗ | ✓ |
| SMAN-BENCH | 12,856 | EN&CN | 7.28 | ∼48k | Both | ✗ | ✓ | ✓ |

unstable. Meanwhile, task evaluation often depends on the process rather than the final page alone, as two failed tasks may represent very different completion progress, making a purely outcome-based comparison unfair across different agents. (2) **Offline evaluation** uses static datasets where the golden path is pre-executed on the device, with actions and screenshots saved offline. The agent generates the current action based on each step's GUI, instructions, and action history, and the action trajectory is formulated as single-path. Although offline benchmarks (Chai et al., 2024; Cheng et al., 2024; Rawles et al., 2023) are more practical for training, considering the diversity of agent task solutions, the agent's good performance may only represent a good fit to the preferences encoded in the current benchmark annotations instead of handling multi-path solutions. This limitation causes some agents (Li et al., 2024b) that achieve exceptionally high performance on certain benchmarks to perform poorly in real-world scenarios, while also exhibiting overly simplistic decision-making. More critically, benchmarks such as MobileAgentBench (Wang et al., 2024c) and AutoDroid (Wen et al., 2024) are constructed on real devices and evaluated within Google apps using the Android Accessibility Service; these apps feature overly clean pages without task-irrelevant ads, buttons, and pop-ups. In real-world scenarios, users may not be able to provide such precise and full instructions all at once (Wang et al., 2024f).

To address the above problems, we introduce a new benchmark named SMAN-Bench with the following data and methods: (1) **Instruction Annotation Method**: To connect one instruction with several trajectories, we use the GIAS to construct 12k instructions with the unlabeled action sequences based on the random walk graph-structured corpus named Mobile3M (Wu et al., 2024a). Each instruction is generated via task templates and slot information based on the corresponding trajectories. (2) **Offline Multi-path Evaluation**: Combining the advantages of both online and offline environments, we propose a multi-path evaluation approach. We allow the agent to execute in a single-path manner and compare the result with the golden path. Alternatively, the agent is also allowed to perform action search within the graph corpus and accumulate step rewards, as it does during online evaluation. (3) **Realistic Noisy Environment**: To explore the effect of noise, we collect an additional sub-dataset named SMAN-Bench-Noisy from apps that are heavily contaminated by advertisement noise. Several apps with substantial ads and pop-ups are specifically selected, including static pop-ups, dynamic video ads, and redirecting advertisement links. Additionally, we also contaminate AITZ Zhang et al. (2024b) by inserting ads into original normal trajectories to build AITZ-Noise. (4) **Active Interactive Evaluation**: We also construct a sub-dataset named SMAN-Bench-Ambiguous, which allows agents to ask when necessary during task execution. Full instructions are pre-constructed and then progressively simplified into ambiguous instructions through slot-based extraction. Questions and answers are built based on slot information and then assigned to the corresponding GUIs.

Our evaluation of general-purpose multimodal models is conducted on three agent frameworks: AppAgent-v1 (Li et al., 2024c), Mobile-Agent-v2 (Ding, 2024), and Mobile-Agent-E (Wang et al., 2025c). Moreover, we include mobile-domain agents trained with continual pre-training, SFT, and RL (Lu et al., 2025a; Luo et al., 2025; Lu et al., 2024; Qin et al., 2025). Finally, we place special emphasis on examining how slow-thinking multimodal models perform in mobile scenarios, analyzing whether this reasoning pattern is effective. Overall, our work makes four main contributions:

• We construct a cross-system benchmark named SMAN-Bench based on Mobile3M's graph structure corpus and propose a slot-based instruction generation method named GIAS.

• We propose an offline multi-path evaluation method and leverage slot-based key node annotations to enable stable assessment of step rewards.

• We introduce SMAN-Bench-Noisy to support realistic noisy evaluation by collecting data from noisy apps, enabling robust assessment under challenging environments.

• We propose SMAN-Bench-Ambiguous to facilitate active interactive evaluation, where agents are allowed to ask clarification questions during execution.

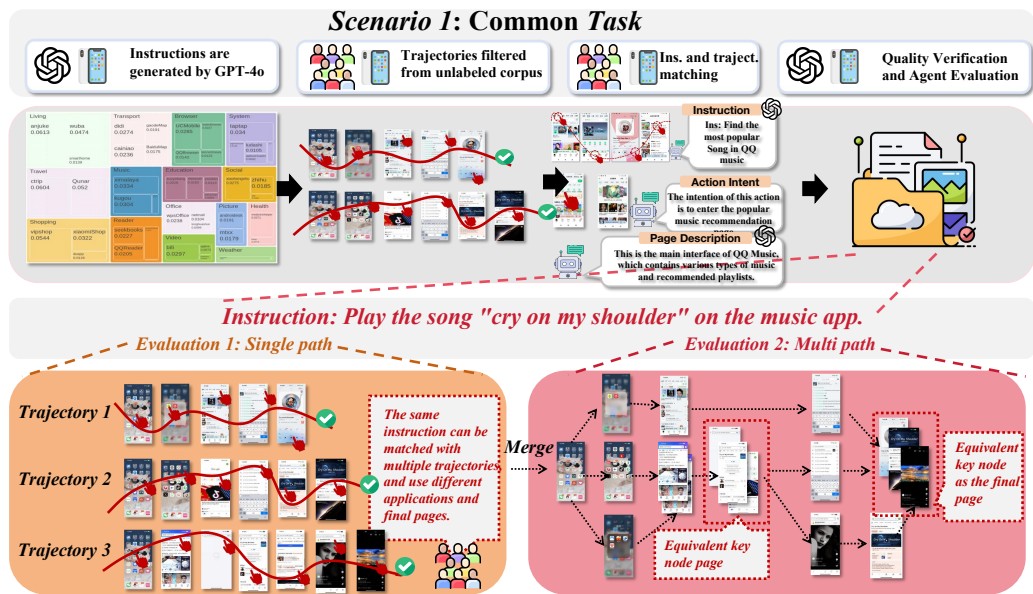

Figure 1: The overview of **SMAN-Bench**, the entire pipeline framework for data construction and filtering, and the distinction between single-path and multi-path evaluation methods.

## 2 RELATED WORK

### 2.1 MOBILE AGENTS

Large language models (Achiam et al., 2023) have emerged as autonomous agents (Li et al., 2024b; Wen et al., 2023) in the mobile domain and garnered considerable attention. With the rapid development of vision-language models (VLMs), multimodal researchers have built mobile GUI agents (Yang et al., 2023; Zheng et al., 2024) and multi-agent frameworks (Ding, 2024; Li et al., 2024c; Wang et al., 2024b) based on closed-source VLMs. Meanwhile, some researchers focus on training agents with stronger element grounding (Cheng et al., 2024; Hong et al., 2024; Wu et al., 2024b), page navigation (Niu et al., 2024; Lu et al., 2024; Gou et al., 2024), GUI understanding (Chai et al., 2024; You et al., 2024; Baechler et al., 2024) and task planning capabilities (Zhang et al., 2024c; Nong et al., 2024; Xu et al., 2024b) based on open-source VLMs. In addition, Digirl (Bai et al., 2024) and Distrl (Wang et al., 2024e) use joint online and offline reinforcement learning to enhance the generalization of mobile agents and mitigate performance degradation when facing app updates and unseen apps. Some researchers (Qinghong Lin et al., 2024) explore optimizing VLM structures. For example, Dorka (Dorka et al., 2024; Tan et al., 2025; Liu et al., 2025; Sun et al., 2024; Xu et al., 2025b; Huang et al., 2025a) optimizes the encoder by incorporating historical images and actions as input.

### 2.2 MOBILE AGENT BENCHMARKS

As shown in Table 1, AndroidEnv (Toyama et al., 2021) and MobilEnv (Zhang et al., 2023) are the first to create LLM agent evaluation environments based on reinforcement learning. Mobile-Bench (Deng et al., 2024) and AppBench (Wang et al., 2024a) introduce online benchmarks combining

API and GUI, while MobileAgentBench (Wang et al., 2024c) establishes the first fully automated multimodal benchmark for VLM-based GUI agents. More offline benchmarks (Li et al., 2020a; Burns et al., 2021; Murthy et al., 2024) are released, which are primarily categorized into GUI understanding and task-oriented. (1) For task-oriented benchmarks, AITW (Rawles et al., 2023) and AITZ (Zhang et al., 2024b) create large-scale benchmarks based on Google apps, while AMEX (Chai et al., 2024) supplements these benchmarks by adding data for GUI understanding with similar app types. ScreenSpot (Cheng et al., 2024), Mobile3M (Wu et al., 2024a), and GUIOdyssey (Lu et al., 2024) focus on more granular element grounding and task planning. (2) Rico (Deka et al., 2017) is the first non-annotated GUI corpus, followed by ScreenQA (Hsiao et al., 2022), Widget Caption (Li et al., 2020b), and Screen2words (Wang et al., 2021), which support Q&A, widget understanding, and page summarization. Subsequently, Mind2web (Deng et al., 2023) incorporates additional GUI data of varying sizes, and Meta-GUI (Sun et al., 2022) provides tasks for multi-round dialogues. Recently, more online benchmarks (Xu et al., 2024a; Rawles et al., 2024; Chen et al., 2024a; Wang et al., 2025a; Huang et al., 2025b; Xu et al., 2025c) have been released to evaluate agents in real environments, such as smartphones, PCs, and web browsers.

# 3  SMAN-BENCH BENCHMARK

## 3.1  MOBILE3M GRAPH-STRUCTURED CORPUS

Mobile3M (Wu et al., 2024a) is a large-scale Chinese&English mobile UI corpus constructed from 49 widely used third-party applications, each with more than ten million active users. The dataset was collected on Android emulators using Appium-based automated interactions and explored with BFS random walk and breadth-first strategies. Each UI page consists of both a screenshot and its corresponding XML document, while interactions such as click, input, and scroll were recorded to build state transitions. In total, the dataset contains 3,098,786 UI pages and approximately 20 million interaction actions, organized into directed graphs, where nodes represent UI pages and edges denote transition actions. To reduce redundancy, a unique-page detection mechanism based on Action Space and pixel differences (BM25) (Robertson et al., 2009) was applied, which improves exploration efficiency and naturally converts the exploration tree into graph structures. The dataset spans a wide range of categories, including travel, lifestyle, and shopping, with relatively balanced distributions, ensuring diversity and representativeness. Unlike prior datasets that only provide isolated pages or chain-structured traces, Mobile3M's graph organization captures the complexity of real-world app interactions, supporting multi-path reasoning and graph-based modeling. More details are described in Appendix C, and the data distribution is shown in Figure 7 and Figure 6.

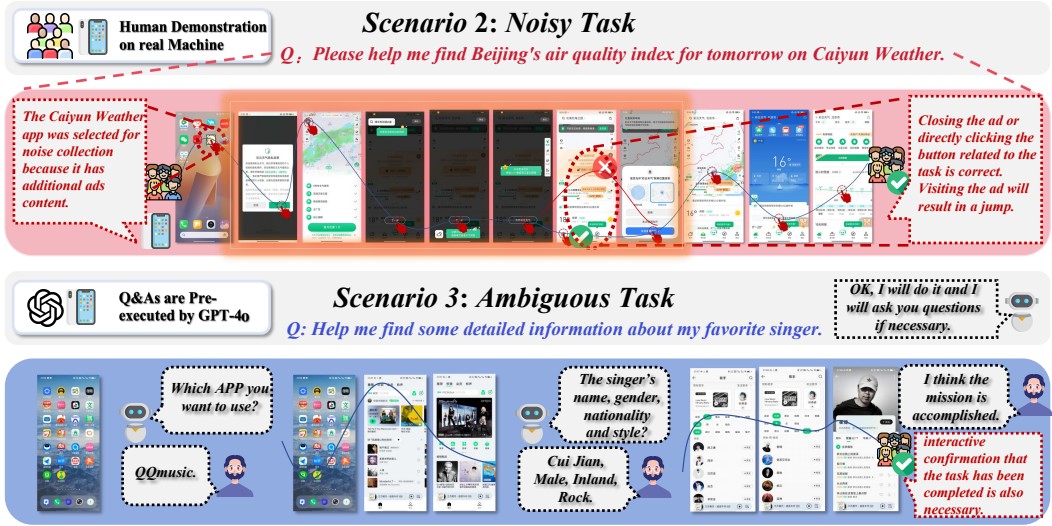

Figure 2: The **SMAN-Bench** includes two other types of tasks: **Noisy-split, and Ambiguous-split**, and demonstrates the process of instruction generation and manual annotation for each task. In Noisy-split, the GUIs with orange shading represent noise.

## 3.2 MULTI-PATH EVALUATION

In the multi-path evaluation setting, a mobile agent can freely explore within the pre-executed and collected graph corpus, provided that the maximum step limit is not exceeded. As shown in Figure 1, several discrete single trajectories are merged into a unified graph, where the merged nodes correspond to the key states of the current instruction. The merging criteria are mainly twofold: (1) Action space and pixel differences (BM25) thresholds (Wu et al., 2024a), which are used to identify the same page (*e.g.*, the main interface of an app may differ slightly across sessions but is considered equivalent). (2) Consistency in button values in the Android XML/Accessibility, where pages are regarded as equivalent if buttons share the same values (Xie et al., 2024) (*e.g.*, searching the same keyword across different browsers produces pages that are merged into one key node). Since pre-execution cannot cover all possible search results, we provide a predefined query pool for instructions, and any search beyond this pool is regarded as invalid. As shown in Figure 2, noisy tasks allow the agent to return to the graph within a limited number of steps. This contrasts with the single-path setting, where advertisements are directly closed, because not all ads are irrelevant to the current task.

## 3.3 DATA CONSTRUCTION

**Generating common instructions from action trajectories.** For the Mobile3M graph corpus, the key challenge is annotating instructions for each trajectory that closely align with the intended actions. Building on Netscape (Murty et al., 2024)'s fine-tuning of web agents to eliminate redundant actions from action sequences, two key points for pairing trajectories and instructions are the **Intent Understanding** and the **Slot Matching** between different GUIs: (i) Using the intent behind actions reduces ambiguity, since coordinate-based actions without GUI pages cannot reproduce scenes accurately. For instance, buttons with the same text view may differ in meaning: in Figure 9, the same "plus" button corresponds to adding "Hazelnut Latte" or "Cookie Mocha". (ii) Filling predefined templates with slot information provides reward signals at key nodes. As GUI agent tasks are inherently multi-path, single-path annotations with unstable preferences can cause performance drops on unseen tasks. Slot filling allows one template to match multiple trajectories that share key nodes, forming the basis of multi-path evaluation. Building on the above findings, we propose a slot-based instruction annotation method named **GIAS** (Generating Instructions From Action Sequences), which is shown in Figure 1. To ensure device diversity, we expanded the evaluation to include additional Android and iOS systems, specifically iOS 18.5, HarmonyOS 5.0, and HyperOS 3.0.

The whole process is as follows: (1) multi-path sampling based on fixed start and end GUIs; (2) GUI content annotation; (3) action intent inference; (4) extracting slot information from GUI changes; (5) filling instruction templates with the slot; (6) deduplication and simplification. The entire process is explained in detail in Algorithm 1. Specifically, we choose paths that start from nodes with the same name in Mobile3M and end at homogeneous nodes with different names (Homogeneous nodes refer to pages whose similarity or the number of identical UI elements exceeds the threshold (Lu et al., 2006)). To ensure trajectory diversity, each trajectory includes at least two different types of actions and minimizes the proportion of intermediate homogeneous pages. Throughout the

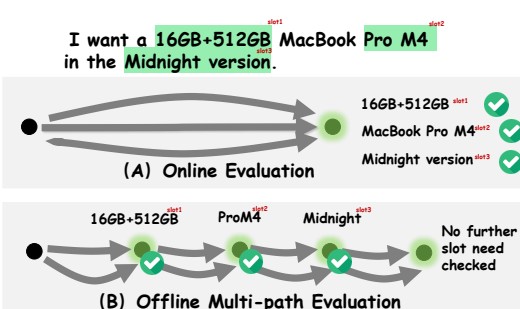

Figure 3: Unlike Online Evaluation, offline multi-path evaluation checks both process and final GUI as reward signals.

entire annotation process, only the verification step involves a closed-source model, while all other steps are performed by open-source models without any human intervention. More details on GIAS are in Appendix D.

**Noisy app and ambiguous instruction data.** SMAN-Bench-Noisy is primarily derived from manual annotation and contamination in existing data: (1) For manual annotation, we select apps from third-party markets; these apps contain unavoidable ads and pop-ups. When performing actions on these apps, we do not handle the following scenarios in advance: login, update, permission settings, ad pop-ups, and VIP subscriptions. In some instructions, to test the agent's response in unexpected

---

**Algorithm 1** GIAS Algorithm

---

**Require:** Start Page, $P_0$; End Page, $P_t$; Trajectory $\sigma$; Page Description $\mathcal{D}$; GUI Pages, $s$; Action, $a$; Action Intent, $T$; Page Slot, $C$; Instruction, $I$; Task, $\mathcal{T}$;
**Ensure:** Prompt, $P$; Few Shot Cases, $F_S$; Verified Flag, $\boldsymbol{Q}$; Instruction Templates, $\gamma$;
 1: Select
$$\sigma_i = \{s_{i_1}, s_{i_2}, \ldots, s_{i_{t-1}}\}, \quad s_{i_j} \sim P_{0:t}, \quad \text{for } j = 1, \ldots, t-1$$
 2: **for** each $s_{ij} \in \sigma_i$ **do**
 3:      **for** $j = 0 : t$ **do**
 4:          $\mathcal{D}_{s_{ij}} \leftarrow \text{VLM}(s_{ij}, P_t)$                       ▷ Get text descriptions from GUIs
 5:          $T_{i_{j:j+1}} \leftarrow \mathcal{D}_{s_{ij}}, \mathcal{D}_{s_{i,j+1}}, a(s_{ij} \rightarrow s_{i,j+1})$      ▷ Get intent between two actions
 6:          $C_{i_{j:j+1}} \leftarrow \mathcal{D}_{s_{ij}}, \mathcal{D}_{s_{i,j+1}}$            ▷ Get Page Slot based on pre-settings
 7:          $I_{ij} \leftarrow (C_{i:t}, T_{i:t}) \sim \text{Uniform}(\gamma)$             ▷ Fill Templates with the slot
 8:      **end for**
 9: **end for**
10: **for** each pair of instructions $(I_i, I_j)$ in $\mathcal{I}$ **do**
11:      **if** $\text{Sim}(I_i, I_j) \geq \tau$ **then**
12:          Discard $I_j$ from $\mathcal{I}$        ▷ Make sure there are no highly similar instructions
13:      **end if**
14: **end for**
15: $\mathcal{T}, Q \leftarrow \text{Veri}\{\mathcal{I}, P, F_S\}$         ▷ for all Trajectories $\sigma_i$, ensuring no redundant steps
16: **return** $\mathcal{T}$

---

situations, we deliberately click on ad pages incorrectly to see if it can recover from divergent paths. All noisy steps and their redirect pages are additionally marked, making this subset adaptable for agents with a rollback mechanism as well. (2) For data contamination, we randomly insert at least one ad (randomly collected from the Google Store app) into AITZ (Zhang et al., 2024b) trajectories, which is a high-quality subset of AITW (Rawles et al., 2023). For SMAN-Bench-Ambiguous, we first construct the full instructions, annotate action trajectories, and remove slot information to build ambiguous instructions. Multiple sets of interactive Q&A are assigned to corresponding GUIs. For example, as shown in Figure 3, the full instruction is: '*I want a 16GB + 512GB MacBook Pro M4 in the Midnight version.*', while the ambiguous instruction is: '*I want to buy a MacBook.*'. Information such as "16GB + 512GB", "Pro M4", and "Midnight version" is treated as three slots, each assigned to the corresponding GUI for step reward. More details can be seen in Appendix E.7.

## 3.4 DATA STATISTICS

The apps and categories in SMAN-Bench remain consistent with Mobile3M, comprising 15 categories and 49 apps, with each category containing at least the top three apps by download volume. As shown on the left part of Figure 4, QQMusic and Kugou account for over 70% of the monthly downloads in the music app category, which are selected as representative music apps in our dataset. The common split includes 12,854 instructions and 800 templates generated by GIAS, which are divided by action steps: simple tasks (1-6 steps) and complex tasks (7-15 steps). As shown in the middle part of Figure 4, there are 9,620 simple tasks with an average of 5.62 steps and 3,234 complex tasks with an average of 8.21 steps. Figure 12 shows the task distribution. We strive for a difficulty balance, but some apps, like *Netmail* (an email tool), still have an imbalance because sending an email often entails multiple steps to complete essential information fields, resulting in a relatively long interaction process that cannot be easily classified as a simple task. From a categorical perspective, shopping apps (*DuApp*) have a higher proportion of complex tasks compared to simple tasks. In contrast, since *Baicizhan* features a clean page and straightforward functionality (*vocabulary learning*), it allows constructing task instructions from templates with fewer slots. The noisy and ambiguous splits each contain 100 instructions, while these noisy data come from another 20 highly noisy apps and each trajectory in the ambiguous split includes at least 5 additional manually constructed Q&A. As shown in Table 16, pre-setting Q&A is strictly aligned with the missing slot information. The average trajectory lengths are 12.74 for the noisy tasks and 7.53 for the ambiguous split. Furthermore, we randomly insert one of 150+ ads at a step within one of the 2,504 trajectories in AITZ. Each trajectory in AITZ-Noisy contains only a single injected advertisement, whereas trajectories in Noisy split include at least five noisy steps, along with additional non-task-related pages such as authorisation, tutorials, redirection pages, and in-app purchase services. More details are in Appendix E.7.

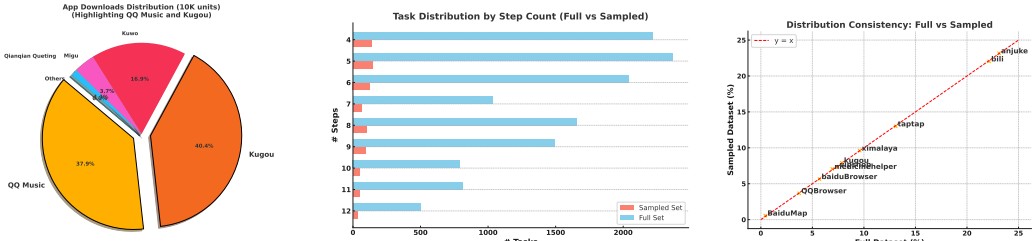

Figure 4: Download volume distribution of music APPs in one month, **on the left**. The distribution of tasks and their respective step counts, **in the middle**. The full distribution of apps and their sampled distribution, **on the right**.

## 4 EXPERIMENT

### 4.1 SETUP

**Models.** For SMAN-Bench common, noisy, and ambiguous splits, we evaluate agent frameworks such as AppAgent-v1, MobileAgent-v2 and MobileAgent-E with different foundation VLMs: InternVL2-40B (Chen et al., 2024b), LLAVA-72B-NEXT (Li et al., 2024a), Qwen2-VL-72B (Wang et al., 2024d), Llama3.2-90B (Dubey et al., 2024), Qwen-VL-Max, GPT-4o (Achiam et al., 2023), and GPT-4v. For open-source mobile agents, we use CogAgent (Hong et al., 2024), UGround-7B (Gou et al., 2024), OS-Atlas-7B (Wu et al., 2024b), UI-Tars (Qin et al., 2025), Kimi-VL (Team et al., 2025), DeepSeek-VL2 (Wu et al., 2024c), OpenCUA-32B (Wang et al., 2025b), GUI-R1 (Luo et al., 2025), UI-R1 (Lu et al., 2025a), UI-S1 (Lu et al., 2025b), and GUI-G1 (Zhou et al., 2025). For reasoning VLMs, we select GLM-4.1v-thinking, Qwen-QVQ-plus, OpenAI o3, Claude-3.7-Sonnet, DeepSeek-R1(HTML) and Doubao-Thinking-pro (Guo et al., 2025).

**Settings.** To reduce cost, only zero-shot evaluations are done on a subset split of *Random-800*, which has a distribution similar to the full split. As shown in Figure 4, the distribution of step and app categories in *Random-800* is fully consistent with that of the full dataset, and 800 instructions correspond one-to-one to 800 predefined instruction templates. For the simple and complex splits, we set the maximum number of steps to 20 and 25, respectively. In ambiguous tasks, direct queries about the next-step decision or the specific function of a button are rejected; only supplementary information relevant to the instruction is allowed to be provided. Unlike AppAgent, we annotate the widget types using specific numbers (Figure 11).

**Metrics.** We follow metrics proposed in MobileAgentBench (Wang et al., 2024c) and AUTO-UI (Zhang & Zhang, 2023),

**Success Rate (SR):** $N_{success}/M_{tasks}$, judged by whether the agent reaches the final pages in multi-path evaluation or whether all actions are correct in single-path evaluation.

**Step Efficiency (SE):** $S_{actual}/S_{min}$, where $S_{actual}$ is the number of actual steps to complete a task, and $S_{min}$ is the task's minimal annotated steps. This metric expresses whether the agent performs unnecessary or redundant actions in multi-path evaluation.

**Step Accuracy (Step.Acc):** $S_{tp}/S_{gt}$, where $S_{tp}$ is the number of predicted actions that match the golden actions in single-path evaluation. This metric also reflects the step reward when the actions are compared with the key nodes in multi-path evaluation.

**TYPE Accuracy:** $S_{ttp}/S_{gt}$, where $S_{ttp}$ is the number of predicted actions that match the type of golden actions. We use TYPE to check whether the action types are correct.

### 4.2 DATA QUALITY VERIFICATION

Due to the random walk involved in the former, the instruction trajectories may include redundant actions, whereas the latter requires checking whether the noise is handled correctly and whether the quality of the Q&A is sufficient. Therefore, we design a data quality verification experiment shown in Table 8 and Table 9. Only 8% of the data constructed in the complex split are found to be suboptimal. This is primarily due to more slots in complex instruction templates, which results in

Table 2: General VLMs with mobile agent framework results on SMAN-Bench Common, Noisy and Ambiguous splits. Type is used in the single-path, while SE is used in the multi-path evaluation.

| Models | Cate. | Common-Simple Type↑/SE↓ | Step. Acc | SR | Common-Complex Type↑/SE↓ | Step. Acc | SR | Noisy Data Type↑/SE↓ | Step. Acc | SR | Ambiguous Data Type↑/SE↓ | Step. Acc | SR |
|---|---|---|---|---|---|---|---|---|---|---|---|---|---|
| _Single-Agent Framework: AppAgent-v1_ | | | | | | | | | | | | | |
| InternVL2-40B | Single | 82.4 | 35.7 | 1.0 | 85.1 | 38.4 | 1.5 | 60.4 | 12.7 | 0.0 | 76.7 | 29.0 | 1.0 |
| | Multi | 5.8 | 43.0 | 10.1 | 4.4 | 46.0 | 6.5 | - | - | - | - | - | - |
| LLAVA-72B-NEXT | Single | 93.2 | 7.4 | 0.0 | 87.7 | 7.8 | 0.0 | 70.4 | 2.7 | 0.0 | 82.3 | 4.8 | 0.0 |
| | Multi | 6.2 | 1.9 | 0.0 | 4.5 | 5.1 | 0.0 | - | - | - | - | - | - |
| Qwen2-VL-72B | Single | 95.2 | 60.3 | 21.1 | 93.5 | 53.8 | 5.0 | 78.0 | 24.4 | 3.0 | 91.2 | 43.5 | 8.0 |
| | Multi | 5.2 | 62.8 | 20.6 | 4.4 | 58.9 | 7.0 | - | - | - | - | - | - |
| Qwen-VL-Max | Single | 94.7 | 58.6 | 20.5 | 91.2 | 54.7 | 7.5 | 77.1 | 24.3 | 3.0 | 90.3 | 48.5 | 9.0 |
| | Multi | 5.9 | 67.6 | 12.6 | 4.3 | 63.1 | 9.6 | - | - | - | - | - | - |
| Llama3.2-VL-90B | Single | 86.4 | 22.4 | 2.6 | 87.0 | 24.3 | 1.0 | 69.7 | 11.2 | 0.0 | 85.4 | 15.5 | 0.0 |
| GPT-4v | Single | 91.2 | 24.0 | 6.0 | 90.8 | 25.2 | 1.0 | 72.7 | 17.4 | 0.0 | 88.6 | 20.5 | 1.0 |
| | Multi | 6.1 | 29.7 | 3.0 | 4.5 | 29.4 | 1.5 | - | - | - | - | - | - |
| GPT-4o | Single | 80.4 | 57.6 | 18.5 | 79.2 | 50.6 | 11.5 | 72.3 | 18.2 | 1.0 | 94.7 | 33.9 | 6.0 |
| | Multi | 5.3 | 61.8 | 19.8 | 4.4 | 61.7 | 16.5 | - | - | - | - | - | - |
| _Multi-Agents Framework: MobileAgent-v2_ | | | | | | | | | | | | | |
| InternVL2-40B | Single | 84.5 | 19.3 | 0.0 | 80.7 | 26.4 | 0.5 | 64.3 | 9.1 | 0.0 | 76.0 | 16.3 | 0.0 |
| | Multi | 6.0 | 27.6 | 3.5 | 4.4 | 32.6 | 3.5 | - | - | - | - | - | - |
| Qwen2-VL-72B | Single | 91.5 | 50.5 | 13.0 | 91.6 | 49.0 | 4.5 | 75.8 | 20.7 | 1.0 | 86.2 | 40.8 | 7.0 |
| | Multi | 5.4 | 54.9 | 15.1 | 4.4 | 58.6 | 8.0 | - | - | - | - | - | - |
| Qwen-VL-Max | Single | 74.2 | 17.0 | 3.0 | 68.8 | 12.3 | 2.0 | 66.5 | 4.2 | 0.0 | 66.6 | 9.0 | 0.0 |
| | Multi | 5.4 | 29.6 | 4.5 | 4.3 | 24.8 | 3.0 | - | - | - | - | - | - |
| Llama3.2-VL-90B | Single | 62.4 | 16.6 | 1.0 | 67.0 | 17.5 | 0.0 | 63.7 | 9.7 | 0.0 | 64.3 | 8.3 | 0.0 |
| GPT-4v | Single | 90.8 | 22.9 | 3.8 | 90.6 | 28.3 | 0.5 | 62.5 | 12.6 | 0.0 | 91.0 | 15.6 | 0.0 |
| | Multi | 6.0 | 17.8 | 5.4 | 4.5 | 11.8 | 0.0 | - | - | - | - | - | - |
| GPT-4o | Single | 91.9 | 53.5 | 13.5 | 92.3 | 50.5 | 17.0 | 77.1 | 25.5 | 2.0 | 91.6 | 39.7 | 12.0 |
| | Multi | 4.9 | 57.6 | 25.5 | 4.2 | 56.3 | 17.5 | - | - | - | - | - | - |
| _Multi-Agents Framework: MobileAgent-E_ | | | | | | | | | | | | | |
| InternVL2-40B | Single | 87.2 | 37.4 | 2.0 | 93.4 | 39.1 | 2.5 | 80.3 | 23.1 | 4.0 | 87.0 | 32.6 | 7.0 |
| | Multi | 5.5 | 46.2 | 9.5 | 4.2 | 46.7 | 8.5 | - | - | - | - | - | - |
| Qwen2-VL-72B | Single | 92.3 | 62.5 | 23.0 | 93.2 | 61.4 | 15.0 | 88.9 | 32.7 | 7.0 | 91.0 | 58.8 | 15.0 |
| | Multi | 4.8 | 60.2 | 18.4 | 4.2 | 63.8 | 12.0 | - | - | - | - | - | - |
| Qwen-VL-Max | Single | 90.6 | 71.5 | 32.5 | 94.8 | 66.3 | 25.5 | 86.2 | 61.6 | 21.0 | 96.3 | 62.5 | 29.0 |
| | Multi | 3.7 | 77.1 | 29.5 | 3.1 | 78.8 | 33.5 | - | - | - | - | - | - |
| Llama3.2-VL-90B | Single | 78.2 | 29.8 | 4.0 | 87.2 | 32.4 | 6.5 | 80.7 | 22.1 | 3.0 | 79.8 | 23.4 | 2.0 |
| GPT-4o | Single | 94.8 | 70.3 | 27.5 | 96.4 | 68.4 | 19.0 | 81.2 | 54.8 | 14.0 | 92.6 | 53.4 | 24.0 |
| | Multi | 3.9 | 77.7 | 30.5 | 3.8 | 70.4 | 26.5 | - | - | - | - | - | - |

unnatural semantics after filling. Semantic alignment does not influence invalid steps between the instructions and the actual trajectories. Therefore, we only manually revised these instructions to ensure that the _random-800_ subset meets the quality requirements.

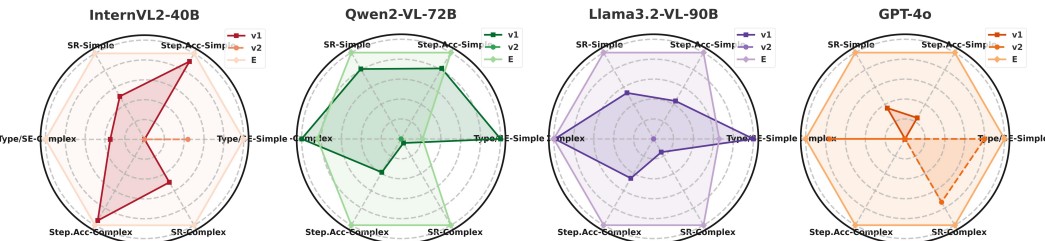

Figure 5: Performance on different backbones and agent frameworks.

## 4.3 MAIN RESULT

**Agent frameworks.** As shown in Figure 5 and Table 2, AppAgent-v1 performs best in single-path evaluations, whereas MobileAgent-v2 excels in multi-path scenarios. This is because single-path evaluation presets the correct historical action and guides agents to focus on the current page. In multi-path, agents' actions depend on their previous decision and pages, so agents are more prone to being trapped in mistaken execution trajectories. MobileAgent-v2's reflection and expectation mechanism effectively mitigates this problem. With dynamic knowledge injection and planning, Mobile-Agent-E surpasses the other two frameworks on most metrics, though models with shorter context windows (_e.g._, Llama3.2-VL-90B) show performance drops due to limited in-context learning.

**Backbones.** As shown in Table 2, the Qwen and GPT series models show superior performance in common tasks, while others perform relatively poorly (50-70% compared to 20-40%). Due to the weaker instruction-following and grounding capabilities of smaller models, their overall performance is limited. The differences in planning ability across models are relatively minor, as reflected in the comparable TYPE accuracy. For one model, the multi-path SR is generally higher than the single-path counterpart, further indicating that multi-path evaluation better reflects the model's true capability.

Table 3: Results on SMAN-Bench Common, Noisy, and Ambiguous splits. Agents include three categories: Continuous pre-training mobile agents, RL-based mobile agents, and reasoning agents.

| Models | Cate. | Common-Simple | | | Common-Complex | | | Noisy Data | | | Ambiguous Data | | |
|---|---|---|---|---|---|---|---|---|---|---|---|---|---|
| | | Type | Step. Acc | SR | Type | Step. Acc | SR | Type | Step. Acc | SR | Type | Step. Acc | SR |
| Continuous Pre-training Mobile Agents | | | | | | | | | | | | | |
| CogAgent-18B | Single | 75.6 | 20.9 | 13.0 | 62.3 | 20.8 | 6.0 | 57.2 | 16.2 | 1.0 | 72.4 | 30.6 | 11.0 |
| UGround-7B | Single | 73.0 | 39.5 | 17.0 | 73.8 | 36.0 | 10.0 | 71.2 | 32.8 | 2.0 | 79.9 | 47.0 | 20.0 |
| UI-Tars-7B-dpo | Single | 75.3 | 41.8 | 19.0 | 75.9 | 37.8 | 12.0 | 73.2 | 35.6 | 4.0 | 81.8 | 49.4 | 23.0 |
| OS-Atlas-7B-pro | Single | 82.1 | 51.5 | 28.0 | 83.5 | 50.6 | 18.0 | 81.2 | 45.3 | 2.0 | 86.4 | 52.3 | 24.0 |
| Kimi-VL-A3B | Single | 75.1 | 21.7 | 12.0 | 61.8 | 21.5 | 6.5 | 56.9 | 15.8 | 1.0 | 71.7 | 29.9 | 11.0 |
| DeepSeek-VL2 | Single | 72.6 | 38.8 | 16.0 | 73.1 | 35.2 | 9.5 | 70.5 | 32.1 | 2.0 | 79.2 | 46.1 | 19.0 |
| UI-Tars-72B-dpo | Single | 94.3 | 64.2 | 32.0 | 96.0 | 63.5 | 24.0 | 94.5 | 59.8 | 7.0 | 92.5 | 66.0 | 30.0 |
| GUI-OWL-7B | Single | 94.7 | 71.2 | 37.5 | 96.0 | 68.4 | 28.5 | 90.8 | 52.4 | 6.0 | 93.8 | 67.2 | 31.0 |
| UI-TARS-1.5-7B | Single | 98.2 | 72.2 | **39.0** | 97.1 | 77.5 | 38.5 | **98.0** | 67.3 | 15.0 | 99.0 | 78.4 | 42.0 |
| OpenCUA-32B | Single | 98.0 | 73.1 | **39.0** | 97.4 | 76.2 | 38.0 | 96.0 | 65.8 | 13.5 | 98.2 | **79.2** | 43.0 |
| RL-based Mobile Agents | | | | | | | | | | | | | |
| UI-R1-3B | Single | 76.5 | 42.7 | 18.5 | 74.8 | 39.1 | 10.5 | 72.6 | 33.9 | 6.0 | 80.9 | 47.8 | 21.5 |
| GUI-R1-3B | Single | 77.2 | 40.9 | 20.0 | 76.4 | 38.6 | 11.0 | 74.1 | 36.8 | 5.5 | 82.1 | 48.6 | 22.5 |
| GUI-G1-3B | Single | 82.4 | 50.8 | 25.5 | 84.1 | 52.3 | 19.0 | 80.6 | 48.7 | 16.5 | 85.3 | 54.9 | 23.0 |
| GUI-R1-7B | Single | 92.8 | 62.7 | 30.5 | 94.2 | 61.9 | 22.5 | 92.7 | 58.1 | 5.5 | 90.9 | 64.3 | 28.5 |
| UI-S1-7B | Single | 95.7 | 65.8 | 33.0 | 97.5 | 65.1 | 25.5 | 96.1 | 61.5 | 8.5 | 94.2 | 67.5 | 31.5 |
| Reasoning Mobile Agents | | | | | | | | | | | | | |
| GLM-4.1v-Thinking | Single | 78.4 | 42.8 | 17.5 | 80.7 | 44.6 | 17.0 | 71.3 | 30.2 | 2.5 | 77.8 | 44.9 | 18.0 |
| Qwen-QVQ-plus | Single | 90.7 | 48.8 | 24.5 | 89.9 | 52.8 | 18.0 | 88.1 | 44.4 | 6.5 | 92.2 | 64.4 | 29.0 |
| OpenAI o3-2025-04-16 | Single | 94.2 | 68.1 | 33.0 | 95.5 | 58.2 | 21.5 | 90.7 | 52.2 | 9.0 | 94.9 | 72.3 | 33.5 |
| Claude 3.7 Sonnet | Single | 98.4 | 74.4 | 38.0 | 98.0 | 76.1 | 38.5 | 95.5 | 59.2 | 10.0 | **99.0** | 77.3 | 41.0 |
| Doubao-1.5-Thinking-pro | Single | 98.2 | 75.8 | **39.0** | 98.1 | **77.3** | 38.5 | 95.9 | 60.1 | 14.0 | **99.0** | 78.0 | 41.5 |
| Claude 4.5 Sonnet | Single | **98.9** | **76.2** | **39.0** | **98.5** | 77.1 | **39.0** | **98.5** | **69.2** | **15.5** | **99.0** | 78.3 | **43.0** |

**Mobile Agents.** As shown in Table 3, compared with agent frameworks, different mobile agents—despite variations in their action spaces—exhibit substantially stronger grounding capabilities, leading to superior overall performance relative to purely framework-based approaches. At the same time, their reasoning outputs are more concise, and the average reasoning steps are significantly reduced to roughly one-fifth of the original pipeline. Notably, the RL-based GUI-G1-3B achieves performance comparable to the pretrained OS-Atlas-7B-pro, highlighting the potential of online reinforcement learning for improving generalization. General-purpose multimodal thinking models, such as Doubao-1.5-Thinking-pro, perform on par with specialized mobile agents (*e.g.*, OpenCUA-32B). When such thinking models are integrated into the above frameworks, no performance gains are observed, suggesting a potential equivalence between frameworks and inherent thinking patterns.

Table 4: Results on AITZ-Noise. Qwen2-VL and OS-Atlas are evaluated on AITZ and AITZ-Noise.

| Agent | Benchmark | General | | Google App | | Install | | Web Shopping | | Total | |
|---|---|---|---|---|---|---|---|---|---|---|---|
| | | Step. Acc | Noisy | Step. Acc | Noisy | Step. Acc | Noisy | Step. Acc | Noisy | Step. Acc | Noisy |
| Qwen2-VL-7B | Normal | 38.5 | - | 44.8 | - | 60.0 | - | 45.1 | - | 46.9 | - |
| | Noise | 37.3 | 15.4 | 42.2 | 17.2 | 54.7 | 20.5 | 42.1 | 17.1 | 43.9 | 17.4 |
| OS-Atlas-7B | Normal | 41.9 | - | 46.4 | - | 60.5 | - | 46.3 | - | 48.6 | - |
| | Noise | 38.8 | 21.8 | 41.7 | 19.7 | 56.4 | 23.5 | 43.8 | 23.6 | 45.1 | 21.7 |

## 4.4 NOISY TASK RESULT

**Out-domain SMAN-Bench-Noisy results.** For noisy data, the results in the third block of Table 2 demonstrate that only a few VLMs complete a task. All VLMs exhibit a declining trend in Step.Acc, particularly LLAVA and Qwen2-VL, while this phenomenon is also observed in open-source agents, due to the absence of noise in their training data. Unlike AITZ-Noise, the ads in Noisy-App are more dynamic and variable, as shown in Figure 16. Specifically, these noises exhibit the following three features: (1) After the pop-up ad countdown ends, the ad disappears automatically, and the agent's delayed instructions may cause accidental taps; (2) Some video ads cannot be closed during the early viewing stages; (3) The mis-taps caused by real ad noise may trigger app redirection.

**Out-domain AITZ-Noise results.** As shown in Table 4, the Step.Acc of Qwen2-VL and OS-Atlas decreased by an average of 3.0% and 3.5% from AITZ normal to in-domain AITZ-Noisy. Given that the Noisy step accuracy is 17.4% and 21.7%, this indicates that open-source agents fail to learn the features of advertisements because a few ads still exist in their training data. They exhibit almost no generalization capability on transferred noisy data, even when only the background screenshot

Table 5: Ablation study on **SMAN-Bench-Ambiguous**.

| Model | AppAgent (Full) | | AppAgent (Ambig.) | | MobileAgent (Full) | | MobileAgent (Ambig.) | |
|---|---|---|---|---|---|---|---|---|
| | Type | StepAcc | Type | StepAcc | Type | StepAcc | Type | StepAcc |
| InternVL2-40B | 70.8 | 21.6 | 76.7 (+5.9) | 29.0 (+7.4) | 61.5 | 8.3 | 76.0 (+15.5) | 16.3 (+8.0) |
| Qwen2-VL-72B | 82.9 | 41.5 | 91.2 (+8.3) | 43.5 (+2.0) | 80.3 | 38.5 | 86.2 (+5.9) | 40.8 (+2.3) |
| Qwen-VL-Max | 81.2 | 39.3 | 90.3 (+9.1) | 48.5 (+9.2) | 57.6 | 3.3 | 66.6 (+9) | 9.0 (+5.7) |
| Llama3.2-VL-90B | 67.9 | 7.6 | 85.4 (+17.5) | 15.5 (+7.9) | 57.8 | 4.0 | 64.3 (+6.5) | 8.3 (+4.3) |
| GPT-4v | 72.8 | 15.9 | 88.6 (+15.8) | 20.5 (+4.6) | 84.6 | 13.9 | 91.0 (+6.4) | 15.6 (+1.7) |
| GPT-4o | 79.7 | 31.9 | 94.7 (+15) | 33.9 (+2) | 87.7 | 38.4 | 91.6 (+3.9) | 39.7 (+1.3) |

changes. When agents become trapped in a page unrelated to the current task, they struggle to determine how to proceed next. More details can be found in Appendix D.2.

### 4.5 AMBIGUOUS INSTRUCTION ABLATION STUDY

As shown in the far-right column of Table 5, all agents exhibit improved performance (by up to 17.5%) when supplied with more informative context through step-by-step Q&A. Ablation results demonstrate that the active interaction module can help agents effectively ignore irrelevant content in task instructions for the current step. This is because full instructions may affect the agent's ability to identify tasks on the current page accurately. In contrast, ambiguous instructions with step-by-step Q&A help the agent better comprehend the page and execute more appropriate actions. However, it is worth noting that not all agents benefit from this mechanism. Weaker VLMs (e.g., InternVL, +5.9%) struggle to generate effective questions, while stronger VLMs (e.g., GPT-4o, +3.9%) are already capable of effective planning and decision-making without additional support. VLMs with intermediate performance, such as LLaMA3.2-VL-90B, benefit more from this mechanism (+17.5%). Meanwhile, end-to-end agents struggle to formulate and explore questions aligned with the predefined settings, highlighting a critical limitation and a promising avenue for future research.

## 5 CONCLUSION

In this paper, we propose SMAN-Bench, a realistic and comprehensive mobile agent benchmark that includes common instruction trajectories, noisy app split, noisy contaminated split, and ambiguous instruction split. We also propose a novel slot-based trajectory annotation method without human evaluation, named GIAS, and an offline multi-path evaluation method, which can assess the agent's ability to obtain step rewards more accurately. This benchmark provides a foundation for evaluating and optimizing GUI agent studies focused on searching for multi-path solutions, noise robustness, and proactive interaction.

ETHICS STATEMENT

We have rigorously refined our dataset to remove any elements that could compromise personal privacy, thereby guaranteeing the highest level of protection for individual data. All data annotations were completed by crowdsourced volunteers, to whom we paid $0.5 per step as compensation and provided the necessary training. The human evaluation of our work was carried out through a meticulously randomized selection of IT professionals. This process ensured a gender-balanced and educationally diverse panel, reflecting a wide spectrum of perspectives and expertise.

REPRODUCIBILITY STATEMENT

All evaluation code, prompts, and datasets used in this paper are open-source. The code and data are mounted and stored on GitHub and HuggingFace platforms, respectively. The full experimental setup is detailed in Section 4.1 and Appendix E. Unless noted, all experiments use the same settings. Overall, these practices make our results reproducible.

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

## A    MOBILE TASK FORMULATION

For mobile agents, there are four essential capabilities: (i) **Overall Planning** to determine the action step sequences. (ii) **Action Thought** to produce an action description at each step (*e.g.,* "open the flight detail page"), (iii) **Element Grounding** to identify a widget (*e.g.,* "[Click]$(x_1, y_1)$") on the GUI, (iv) **Action Reflection** to determine whether the next GUI matches the expectation.

$$\hat{a}_t = \begin{cases} [x_1, y_1, x_2, y_2], & \hat{a}_t \in \text{Click} \\ \{\uparrow, \downarrow, \leftarrow, \rightarrow\}, & \hat{a}_t \in \text{Scroll} \\ text, & \hat{a}_t \in \text{Type} \end{cases} \tag{1}$$

Given a mobile screenshot $\mathcal{S}$ (*e.g.,* a Ctrip screenshot on Android) and a task $\mathcal{T}$ (*e.g., "Book a flight ticket from Chengdu to Beijing Sep.15 for me."*), a GUI agent should generate a sequence of executable actions. Specifically, at time step $t$, the agent should select an action $\boldsymbol{a}_t$ from the action space $\mathcal{A}$, which includes three types of actions: (1) Click. (2) Scroll. (3) Input. Based on the current environment observation $\mathcal{S}_t$, the action history $\mathcal{H}_{1:t-1} = \{\hat{a}_1, \hat{a}_2, ..., \hat{a}_{t-1}\}$, and the last step reflection $\boldsymbol{f}_{t-1}$, the GUI agent will generate plan $\mathcal{P}_t$:

$$\mathcal{P}_t = \left\{ \hat{a}_t^{(1)} \cdots \hat{a}_t^{(n)} \mid (\hat{a}_1, \cdots, \hat{s}_{t-1}), f_{t-1}, \mathcal{S}_t \right\} \tag{2}$$

where $\mathcal{P}_t$ represents the planning of the next $n$ actions starting from the current step. The environment observation $\mathcal{S}_t$ comprises an HTML document $text_t$ and a mobile screenshot $image_t$.

## B    THE FULL RELATED WORKS

### B.1    MOBILE AGENTS

Large language models (Achiam et al., 2023) have emerged as autonomous agents (Li et al., 2024b; Wen et al., 2023) in the mobile domain and garnered considerable attention. With the rapid development of vision-language models (VLMs), multimodal researchers have built mobile GUI agents

(Yang et al., 2023; Zheng et al., 2024) and multi-agent frameworks (Ding, 2024; Li et al., 2024c; Wang et al., 2024b) based on closed-source VLMs. Meanwhile, some researchers focus on training agents with stronger element grounding (Cheng et al., 2024; Hong et al., 2024; Wu et al., 2024b), page navigation (Niu et al., 2024; Lu et al., 2024; Gou et al., 2024), GUI understanding (Chai et al., 2024; You et al., 2024; Baechler et al., 2024) and task planning capabilities (Zhang et al., 2024c; Nong et al., 2024; Xu et al., 2024b) based on open-source VLMs. In addition, Digirl (Bai et al., 2024) and Distrl (Wang et al., 2024e) use joint online and offline reinforcement learning to enhance the generalization of mobile agents and mitigate performance degradation when facing app updates and unseen apps. Some researchers (Qinghong Lin et al., 2024) explore optimizing VLM structures. For example, Dorka (Dorka et al., 2024) optimizes the encoder by incorporating historical images and actions as input.

### B.2 OFFLINE MOBILE AGENT BENCHMARKS

As shown in Table 1, AndroidEnv (Toyama et al., 2021) and MobilEnv (Zhang et al., 2023) are the first to create LLM agent evaluation environments based on reinforcement learning. Mobile-Bench (Deng et al., 2024) and AppBench (Wang et al., 2024a) introduce online benchmarks combining API and GUI, while MobileAgentBench (Wang et al., 2024c) establishes the first fully automated multimodal benchmark for VLM-based GUI agents. More offline benchmarks (Li et al., 2020a; Burns et al., 2021; Murthy et al., 2024) are released, which are primarily categorized into GUI understanding and task-oriented. (1) For task-oriented benchmarks, AITW (Rawles et al., 2023) and AITZ (Zhang et al., 2024b) create large-scale benchmarks based on Google apps, while AMEX (Chai et al., 2024) supplements these benchmarks by adding data for GUI understanding with similar app types. ScreenSpot (Cheng et al., 2024), Mobile3M (Wu et al., 2024a), and GUIOdyssey (Lu et al., 2024) focus on more granular element grounding and task planning. (2) Rico (Deka et al., 2017) is the first non-annotated GUI corpus, followed by ScreenQA (Hsiao et al., 2022), Widget Caption (Li et al., 2020b), and Screen2words (Wang et al., 2021), which is for Q&A, widget understanding, and page summarization. Subsequently, Mind2web (Deng et al., 2023) incorporates additional GUI data of varying sizes, and Meta-GUI (Sun et al., 2022) provides tasks for multi-round dialogues.

### B.3 ONLINE MOBILE AGENT BENCHMARKS

Recently, more online benchmarks (Xu et al., 2024a; Rawles et al., 2024; Chen et al., 2024a; Wang et al., 2025a; Huang et al., 2025b) have been released to evaluate agents in real environments, such as smartphones, PCs, and web browsers. Existing OS-level benchmarks, such as OSWorld (Xie et al., 2024), focus on open-ended, long-horizon computer-use tasks on desktop operating systems. AndroidWorld (Rawles et al., 2024), by contrast, provides 20 applications and 116 programmatically generated daily tasks within a controlled Android emulator, along with structured reward signals, enabling reproducible evaluation of mobile GUI agents. AndroidLab (Xu et al., 2025d) further extends this line of work by operating in real or near-real Android environments and constructing large families of tasks from composable atomic skills, thereby systematically assessing an agent's generalization across task compositions and across applications.

### B.4 GRAPH-BASED GUI AGENT BENCHMARKS

Recent efforts in evaluating GUI-based and web-oriented agents have produced several benchmarks that differ substantially in task domains, instruction sources, graph-structured trajectory modeling, and reward assignment strategies. ColorBench (Liang et al., 2025) focuses on long-horizon, cross-app search and query tasks driven by manually written instructions. It adopts VLM-based page-content descriptions and embeds them to compute similarity scores for node merging, while rewarding agents through sub-task verification of necessary actions in both online and mixed evaluation environments. WebGraphEval (Qian et al., 2025), in contrast, targets shopping and web-retrieval scenarios within WebArena, where multiple

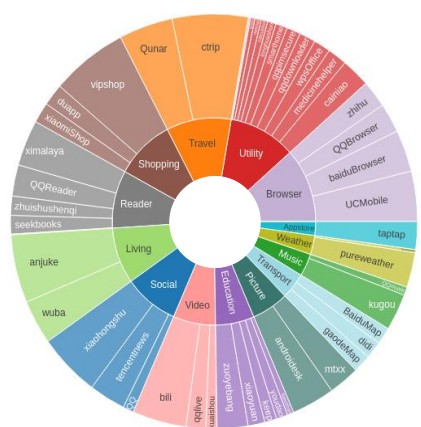

Figure 6: Categories and apps used in the Mobile3M Dataset

| Benchmark | Task Type | Instruction Source | Graph Node Merging Strategy | Key-Node Rewarding |
|---|---|---|---|---|
| ColorBench | Search & query; cross-app long-horizon tasks | Manually written instructions | VLM-based page description; embedding similarity used to merge nodes | Sub-task decomposition; verifying necessary actions for reward |
| WebGraph | Shopping & retrieval in web tasks | WebArena tasks | Multiple agent trajectories merged by action-sequence similarity | Final-answer correctness back-propagated to earlier steps |
| CRAB | Calendar, map, web | Reverse DAG decomposition to generate instructions | Cross-device node alignment | Sub-task decomposition; reward via checking environment state |
| OmniBench | Office, video editing, web | Reverse DAG decomposition | Trajectory-node merging for instruction generation | None; additional intent signals ensure data quality |
| SMAN-Bench (ours) | Everyday tasks: music, travel, search, shopping; more app categories | Online log filtering + GIAS-based synthesis | Action-space validity checking + page-content clustering | Slot-based matching to locate correct action and downstream key-node page; rewarding equivalent actions |

Table 6: Comparison of SMAN-Bench with existing graph-based and task-oriented GUI agent benchmarks.

agent trajectories for the same task are aligned and merged by comparing action-sequence similarity. Rewards for intermediate steps are back-propagated solely from the correctness of the final answer, enabling offline evaluation enhanced with LLM-as-judge. CRAB (Xu et al., 2025a) extends to heterogeneous environments such as calendars, maps, and general web interfaces; its instructions are produced through reverse DAG task decomposition, and node alignment is handled across devices. Rewards are obtained by verifying environment states after sub-tasks, and evaluation is conducted through cross-system online interaction. OmniBench (Bu et al., 2025) further broadens the domain to office automation, video editing, and complex web tasks. It also uses reverse DAG decomposition but merges trajectory nodes to generate high-quality instructions, supported by additional intent signals instead of explicit reward modeling. Its evaluation emphasizes sub-task–level assessment. Overall, these benchmarks reveal a diverse design landscape—from instruction construction and trajectory graph fusion to reward propagation and evaluation modes—highlighting the need for more unified and robust frameworks for training and assessing long-horizon multimodal agents.

As shown in Table 6, compared with previous benchmarks, SMAN-Bench (ours) targets more realistic everyday mobile scenarios, including music, travel, search, and shopping—spanning a broader set of apps. Instead of relying on handcrafted tasks or constrained trajectories, we construct instructions through online log filtering and GIAS-based synthesis. For trajectory consolidation, SMAN-Bench combines action-space validity checking with page-content clustering, enabling more reliable merging of semantically equivalent states. Reward assignment is also more fine-grained: slot-based matching automatically identifies the correct action and downstream key-node page, allowing both intermediate rewards and final-answer evaluation in a fully offline setting. These designs make SMAN-Bench a more scalable and realistic benchmark for long-horizon GUI agents.

## C    MOBILE3M DATASET

Mobile3M is a large-scale dataset designed to systematically explore and analyze the functionality of mobile applications through UI-based interactions. It provides a comprehensive representation of user interface (UI) elements, interactions, and app navigation patterns. Mobile3M is characterized by the following key features:

| Category | APP | Unique Nodes | All Nodes | Action Steps | All Nodes(%) | Category | APP | Unique Nodes | All Nodes | Action Steps | All Nodes (%) |
|---|---|---|---|---|---|---|---|---|---|---|---|
| Living | anjuke | 57,286 | 190,102 | 1,334,428 | 6.13% | Reader | seekbooks | 15,902 | 70,266 | 563,882 | 2.27% |
| Living | wuba | 38,667 | 147,009 | 903,586 | 4.74% | Reader | QQReader | 22,588 | 63,458 | 472,509 | 2.05% |
| Living | smarthome | 12,595 | 42,961 | 304,816 | 1.39% | Reader | zhuishushenqi | 14,737 | 63,210 | 392,903 | 2.04% |
| Travel | ctrip | 63,449 | 187,079 | 1,217,304 | 6.04% | Reader | pdfreader | 495 | 1,507 | 5,211 | 0.05% |
| Travel | Qunar | 42,462 | 161,005 | 1,211,015 | 5.20% | Social | xiaohongshu | 45,324 | 85,362 | 525,519 | 2.75% |
| Shopping | vipshop | 72,468 | 168,531 | 1,036,086 | 5.44% | Social | zhihu | 21,766 | 57,261 | 373,756 | 1.85% |
| Shopping | xiaomiShop | 21,666 | 99,770 | 755,718 | 3.22% | Social | QQ | 7,051 | 20,600 | 141,969 | 0.66% |
| Shopping | duapp | 18,925 | 38,926 | 223,379 | 1.26% | Education | zuoyebang | 19,884 | 70,661 | 507,146 | 2.28% |
| Transport | didi | 12,786 | 84,865 | 637,400 | 2.74% | Education | Xiaoyuan | 10,727 | 56,806 | 393,395 | 1.83% |
| Transport | cainiao | 20,593 | 73,132 | 480,223 | 2.36% | Education | Youdao | 8,756 | 35,121 | 206,035 | 1.13% |
| Transport | gaodeMap | 13,674 | 59,142 | 319,377 | 1.91% | Education | Baicizhan | 4,196 | 16,383 | 88,500 | 0.53% |
| Transport | BaiduMap | 13,552 | 54,322 | 280,498 | 1.75% | Office | wpsOffice | 11,156 | 73,739 | 486,661 | 2.38% |
| Browser | UCMobile | 40,618 | 88,220 | 615,049 | 2.85% | Office | Netmail | 5,544 | 32,308 | 260,682 | 1.04% |
| Browser | baiduBrowser | 36,016 | 70,282 | 401,348 | 2.27% | Office | tonghuashun | 6,410 | 30,722 | 163,297 | 0.99% |
| Browser | QQBrowser | 18,500 | 44,006 | 218,828 | 1.42% | Office | QQmail | 712 | 1,590 | 4,597 | 0.05% |
| Browser | tencentnews | 23,408 | 38,241 | 224,804 | 1.23% | Video | bili | 46,080 | 91,891 | 471,940 | 2.97% |
| System | taptap | 24,759 | 105,461 | 624,941 | 3.40% | Video | qqlive | 12,497 | 22,601 | 99,677 | 0.73% |
| System | qqpimsecure | 8,997 | 42,691 | 379,926 | 1.38% | Video | kuaishou | 7,126 | 12,115 | 59,373 | 0.39% |
| System | ludashi | 2,773 | 32,474 | 219,804 | 1.05% | Picture | androidesk | 28,432 | 59,228 | 418,773 | 1.91% |
| System | qqdownloader | 10,517 | 28,502 | 151,824 | 0.92% | Picture | mtxx | 19,718 | 55,324 | 419,055 | 1.79% |
| System | calculator | 4,265 | 15,819 | 97,005 | 0.51% | Health | medicinehelper | 15,046 | 83,832 | 547,880 | 2.71% |
| System | supercaculator | 690 | 1,369 | 5,444 | 0.04% | Health | keep | 7,730 | 22,500 | 117,124 | 0.73% |
| Music | ximalaya | 34,995 | 103,395 | 577,032 | 3.34% | Weather | pureweather | 25,252 | 79,283 | 610,695 | 2.56% |
| Music | kugou | 40,043 | 94,271 | 504,368 | 3.04% | Weather | cloudweather | 1,956 | 3,904 | 19,339 | 0.13% |
| Music | QQmusic | 5,545 | 17,539 | 64,211 | 0.57% | 15 | 49 | 998,334 | 3,098,786 | 20,138,332 | 100% |

Figure 7: The data distribution in Mobile3M.

(1) **Scale and Diversity**
Mobile3M includes over 20 million user interactions, covering 3 million screenshots and corresponding XML documents. These data are organized into directed graphs for 49 widely-used Chinese apps, where nodes represent UI pages, and edges capture user actions.

(2) **Detailed UI Representation**
Each UI page is described by both a screenshot and an XML document. The XML documents provide detailed structural information, including UI elements (e.g., buttons, text fields), their hierarchical relationships, and layout properties such as bounding boxes.

(3) **Action Space**
The dataset defines three fundamental user actions—*click*, *scroll*, and *input*—to simulate real-world app interactions. Each UI page contains an action space derived from its interactable elements, facilitating comprehensive modeling of user behaviors.

(4) **Graph-Based Organization**
Mobile3M employs a breadth-first search (BFS) algorithm to explore app functionality, representing the exploration results as graphs. This structure enables the identification of app workflows, the relationship between UI pages, and the possible transitions triggered by user actions.

(5) **Efficiency and Optimization**
To enhance exploration efficiency, Mobile3M incorporates a "unique page" mechanism that eliminates duplicates by comparing UI pages using a combination of element and pixel-based similarity thresholds. This reduces the exploration space, prevents redundant actions, and avoids cyclic sequences, ensuring more diverse and meaningful data coverage.

(6) **Balanced Action Distribution**
The dataset emphasizes balanced representation of user actions by prioritizing underrepresented interactions, such as *input*. For example, random keywords are introduced for input actions, and scroll actions are executed in multiple directions to capture diverse app behaviors.

(7) **Task-Oriented Exploration**
Inspired by APPAgent, the dataset leverages a random walk algorithm to systematically interact with UI elements and record transitions between pages. The exploration process captures action traces, enabling task-driven navigation and detailed understanding of app functionalities.

# D  DATA ANALYSIS AND CONSTRUCTION

## D.1  GIAS PROMPT

**Input Format**

*You will be provided with a series of user interaction histories, each consisting of a caption describing the current page and an action performed by the user.*

**Your Task**

*Analyze each action and the corresponding page caption to determine what action was taken on that page. Summarize these actions into a task description, which should be a request. For example:*

- *"I want to see what VIP privileges are available."*
- *"Help me find pants on sale."*
- *"Tell me what items are in my shopping cart."*

**Important Notes:**

1. *The task description and the sequence of actions should have a logical relationship.*
2. *The task description should be phrased as a request, reflecting the goal of the actions taken.*
3. *Actions and captions should be analyzed in sequence to deduce the user's objective.*

**Output Format**

**"step-by-step description"**: *"Provide a series of interactions, where each entry corresponds to a screenshot caption of the current phone screen and the action performed on that page."*

**"concise task"**: *"Summarize the user's overall goal based on the step-by-step description."*

**Example**

**Caption 1:**
*This image shows a screenshot of a shopping application interface.*

**Action 1:**
*Click(Skincare Set)*

**Caption 2:**
*This image shows a screenshot of a shopping application interface. At the top, there is a search bar with the text "Skincare Set." Additionally, at the bottom of the page, there is a navigation bar with options like "All Products," "New Arrivals," "Moisturizing," "Dry Skin," "Niacinamide," and "Hyaluronic Acid." The current state is "All Products."*

**Action 2:**
*Click(New Arrivals)*

**Caption 3:**
*This image shows a screenshot of a shopping application interface. At the top, there is a search bar with the text "Skincare Set." Additionally, there is a navigation bar at the bottom with options like "All Products," "New Arrivals," "Moisturizing," "Dry Skin," "Niacinamide," and "Hyaluronic Acid." The current state is "New Arrivals." Below are multiple product recommendations.*

**Action 3:**
*Click(Ad)*

**Caption 4:**
*This image shows a product detail page. At the top, there is a pink banner that reads "Buy a set and get 13 items free," along with a product photo.*

**Output:**

**"step-by-step description":**

1. *Click the "Skincare Set" product under the "Beauty" subcategory of "Recommended."*

2. *On the Skincare Set search results page, click the "New Arrivals" tab.*

3. *On the product details page, click the "Ad" tab.*

**"Concise task":**
*Help me find the latest skincare set that is on promotion.*

**New Input and Task**

*Now, based on the following input, please generate the "step-by-step description" and "concise task":*

```
{trajectory_description}
```

## D.2 DATA CONTAMINATION

The collected advertisements are shown in Figure 10. We embed them into the normal dataset and applied background whitening. We ensure that the elements that should have been clicked on the current page are no longer visible after the contamination. When splitting the training and test data, the position of the embedded advertisements is randomly assigned. However, the types of advertisements in the training data are largely consistent with those in the test data, and the same advertisements maintain consistent embedding positions.

Table 7: Qwen2-VL and OS-Atlas fine-tuned on AITZ-Noise, AITW, or AITZ and evaluation on AITZ-Noise(Out-domain). Metric "Noisy" means out-domain noisy step accuracy.

| Agent | Training Data | General | | Google App | | Install | | Web Shopping | | Total | |
|---|---|---|---|---|---|---|---|---|---|---|---|
| | | Step. Acc | Noisy | Step. Acc | Noisy | Step. Acc | Noisy | Step. Acc | Noisy | Step. Acc | Noisy |
| *Normal Data Supervised Fine-tuning* | | | | | | | | | | | |
| Qwen2-VL-7B | AITZ | 37.33 | 15.38 | 42.18 | 17.11 | 54.68 | 20.45 | 42.06 | 17.14 | 43.84 | 17.46 |
| OS-Atlas | AITZ | 38.81 | 21.79 | 41.61 | 19.73 | 56.37 | 23.57 | 43.71 | 23.57 | 45.16 | 21.62 |
| *In-domain Noise Supervised Fine-tuning* | | | | | | | | | | | |
| Qwen2VL | AITZ + Noisy | 43.07 | 77.56 | 47.63 | 73.68 | 60.64 | 75.76 | 44.02 | 75.00 | 48.20 | 75.79 |
| OS-Atlas | AITZ + Noisy | 44.92 | 82.05 | 49.64 | 76.32 | 63.06 | 79.55 | 48.01 | 78.57 | 50.99 | 79.56 |
| *Out-domain Noise Supervised Fine-tuning* | | | | | | | | | | | |
| Qwen2VL | AITZ + Noisy | 37.18 | 50.64 | 45.18 | 41.89 | 57.45 | 53.38 | 42.32 | 50.00 | 44.96 | 49.90 |
| OS-Atlas | AITZ + Noisy | 41.75 | 53.85 | 45.47 | 48.65 | 60.27 | 60.90 | 47.28 | 55.71 | 48.69 | 55.47 |

# E EXPERIMENT DETAILS

## E.1 BASELINE MODEL DEMONSTRATION

**AppAgent** Below is the prompt we used. We did not re-adapt or adjust the prompt for different base models to ensure fairness.

```
1  I will give you the screenshot of a mobile app, the clickable UI element is labeled
2  with a letter 'c' and the number <ui_element> on the screen. The tag of each element is located
      at the center of the
3  element. Clicking on this UI element is a necessary part of proceeding with a larger task,
      which is to <task_description>.
4  In order to realize this larger task, you must first realize the current task <current_task_desc>
      in current screenshot.
5  Your task is to describe the functionality of the UI element concisely in one or two sentences.
      Notice that your
6  description of the UI element should focus on the general function. For example, if the UI
      element is used to navigate
7  to the chat window with John, your description should not include the name of the specific
      person. Just say:
8  "Clicking this area will navigate the user to the chat window". Never include the tag of the
```

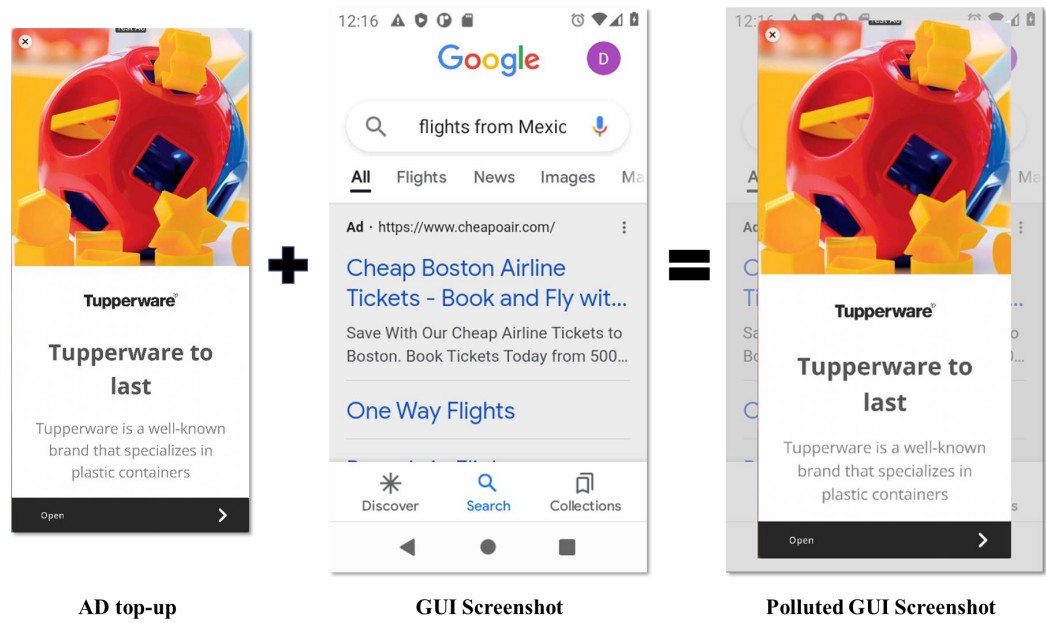

**AD top-up**        **GUI Screenshot**        **Polluted GUI Screenshot**

Figure 8: Contaminated datasets are constructed by inserting advertisements and whitening the background of the original GUI screenshots.

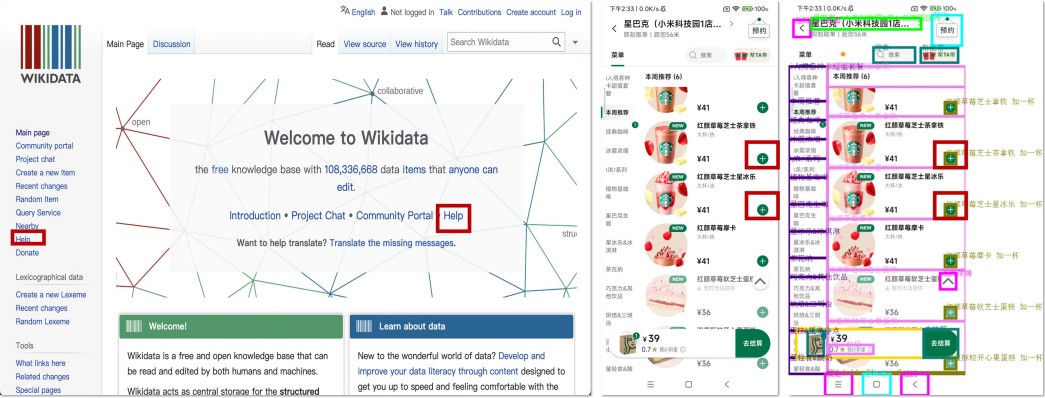

Figure 9: The red boxed areas represent identical graphical controls and identical textual controls, which can create ambiguity in the action history.

```
9  UI element in your description. You can use pronouns such as "the UI element" to refer to the
       element.
```

Listing 1: Click Document Template

```
1  I will give you the screenshot of a mobile app, the clickable UI element is labeled
2  with a letter 'c' and the number <ui_element> on the screen. The tag of each element is located
        at the center of the
3  element. Clicking on this UI element is a necessary part of proceeding with a larger task,
        which is to <task_description>.
4  In order to realize this larger task, you must first realize the current task <current_task_desc>
        in current screenshot.
5  Your task is to describe the functionality of the UI element concisely in one or two sentences.
        Notice that your
```

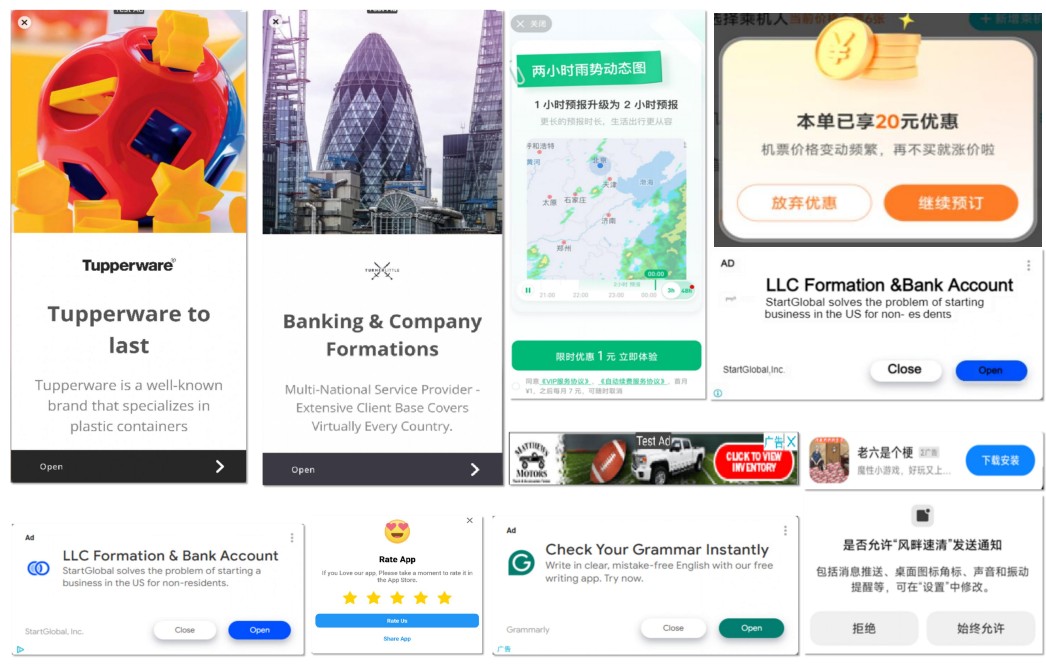

Figure 10: A collection of pop-up ads, collected from Google service official apps, third-party market apps, and mobile apps in mainland China.

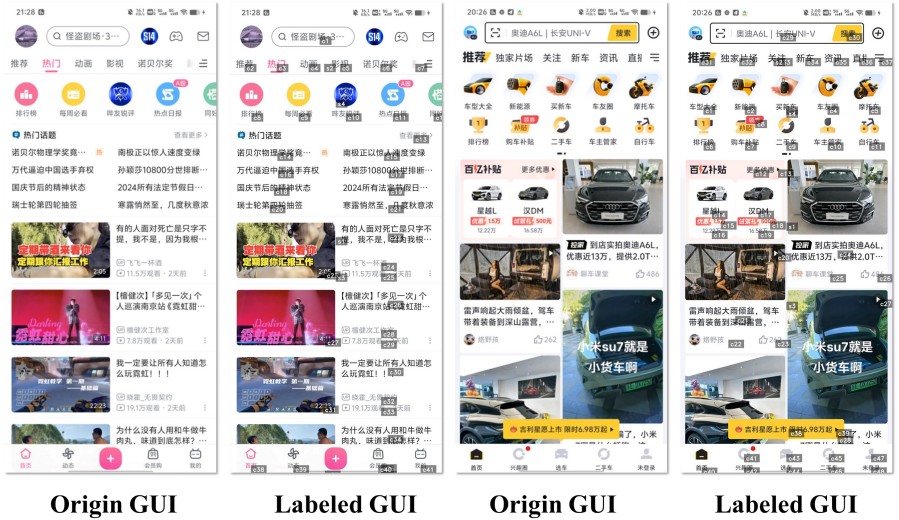

**Origin GUI**     **Labeled GUI**     **Origin GUI**     **Labeled GUI**

Figure 11: AppAgent GUI labeled method.

6   description of the UI element should focus on the general function. For example, if the UI element is used to navigate

```
7  to the chat window with John, your description should not include the name of the specific
       person. Just say:
8  "Clicking this area will navigate the user to the chat window". Never include the tag of the
9  UI element in your description. You can use pronouns such as "the UI element" to refer to the
       element.
```

Listing 2: Click Documentation Template

```
1  A documentation of this UI element generated from previous demos is shown below. Your
2  generated description should be based on this previous doc and optimize it. Notice that it is
       possible that your
3  understanding of the function of the UI element derived from the given screenshots conflicts
       with the previous doc,
4  because the function of a UI element can be flexible. In this case, your generated description
       should combine both.
5  Old documentation of this UI element: <old_doc>
```

Listing 3: Refine Documentation Suffix

```
1  You are an agent that is trained to perform some basic tasks on a smartphone. You will be
       given a
2  smartphone screenshot. The interactive clickable UI elements on the screenshot are labeled
       with tags starting from "c1".
3  The interactive scrollable UI elements on the screenshot are labeled with tags starting from "s1
       ".The tag of each
4  interactive element is located in the center of the element. Every screenshot I've given you is a
       screenshot after
5  executing the correct action.
6
7  You can call the following functions to control the smartphone:
8
9  1. click(element: str)
10 This function is used to click an UI element shown on the smartphone screen.
11 "element" is a tag assigned to an UI element shown on the smartphone screen.
12 A simple use case can be click(c5), which taps the UI element labeled with "c5".
13
14 2. input(text_input: str)
15 This function is used to insert text input in an input field/box. text_input is the string you want
       to insert and must
16 be wrapped with double quotation marks. A simple use case can be text("Hello, world!"),
       which inserts the string
17 "Hello, world!" into the input area on the smartphone screen. This function is usually callable
       when you see a screenshot
18 about text inputing.
19
20 3. scroll(element: str, direction: str)
21 This function is used to scroll an UI element shown on the smartphone screen, usually a scroll
       view or a slide bar.
22 "element" is a tag assigned to an UI element shown on the smartphone screen. "direction" is a
       string that
23 represents one of the four directions: up, down, left, right. "direction" must be wrapped with
       double quotation
24 marks.
25 A simple use case can be swipe(s21, "up"), which scroll up the UI element labeled with "s21".
26
27 <ui_document>
28 The task you need to complete is to <task_description>, to complete this task you should
       perform current task
```

Table 8: Quality verification results. Crowdsourced annotations verified by agent professionals.

| Metric | Simple | Complex | Noisy | Ambiguous |
|---|---|---|---|---|
| Annotation Step | 5.62 | 8.21 | 12.74 | 7.53 |
| Evaluation Step | 5.57 | 8.07 | 12.69 | 7.43 |
| Win Rate↑ | 96.0 | 92.0 | 100.0 | 100.0 |
| SE↓ | 1.01 | 1.02 | 1.00 | 1.01 |

Table 9: App counts and proportions in full and subset datasets.

| App | Full | Subset | Full (%) | Sampled (%) |
|---|---|---|---|---|
| BaiduMap | 52 | 2 | 0.43 | 0.56 |
| QQBrowser | 447 | 13 | 3.72 | 3.67 |
| anjuke | 2773 | 82 | 23.11 | 23.16 |
| baiduBrowser | 682 | 20 | 5.68 | 5.65 |
| bili | 2658 | 78 | 22.15 | 22.03 |
| kugou | 943 | 28 | 7.86 | 7.91 |
| medicinehelper | 831 | 25 | 6.93 | 7.06 |
| taptap | 1570 | 46 | 13.08 | 12.99 |
| vipshop | 893 | 26 | 7.44 | 7.34 |
| ximalaya | 1151 | 34 | 9.59 | 9.60 |

```
29  <current_task_desc>. Your past actions to proceed with this task are summarized as follows: <
        last_act>
30  Now, given the documentation and the following labeled screenshot, you need to think and
        call the function needed to
31  proceed with the task. Your output should include three parts in the given format:
32  Observation: <Describe what you observe in the image>
33  Thought: <To complete the given task, what is the next step I should do>
34  Action: <The function call with the correct parameters to proceed with the task.>
35  Summary: <Summarize your past actions along with your latest action in one or two sentences
        . Do not include the
36  tag in your summary>
37  You can only take one action at a time, so please directly call the function.
```

Listing 4: Task Template

## E.2  SAMPLED DATA QUALITY VERIFICATION

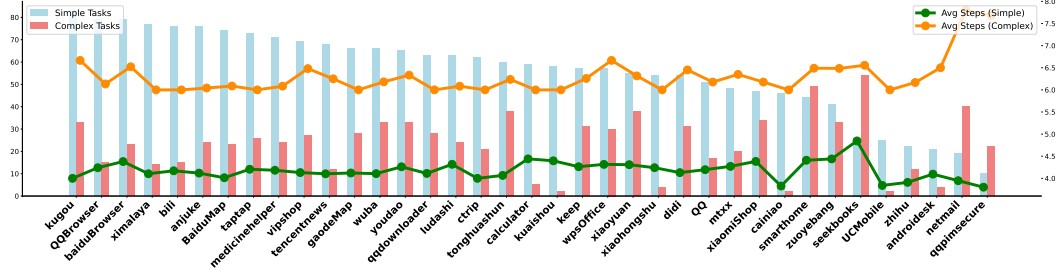

Figure 12: The **task distribution** chart is sorted by the number of simple tasks in descending order. The average steps for both simple and complex tasks in each app remain relatively balanced.

## E.3  SAMPLING REPRESENTATIVENESS

To construct a representative evaluation subset, we perform stratified sampling from the full dataset, with application-level proportions preserved. As detailed in Table 9, the sampled subset closely mirrors the original distribution across a diverse range of applications. Despite a substantial reduction in data volume, the relative proportions of key apps (e.g., anjuke, bili, taptap) remain nearly identical between the full dataset and the sampled subset (e.g., 23.11% vs. 23.16%, 22.15% vs. 22.03%). This strong alignment demonstrates the effectiveness of our sampling strategy in maintaining distributional

| Benchmark | Data Source | Music | Travel | Shopping | Office/Tools |
|---|---|---|---|---|---|
| Mobile-Bench Deng et al. (2024) | Online log filtering | 93 | 92 | 98 | – |
| WebArena Zhou et al. (2023) | Human-written + template-generated | – | – | 88 | 90 |
| AndroidWorld Rawles et al. (2024) | Human-written + template-generated | 90 | 85 | 93 | – |
| SMAN-Bench (ours) | Real logs + template-generated | 92 | 96 | 90 | 93 |

Table 10: Comparison of benchmarks, data sources, and category coverage quality.

fidelity. By preserving both frequent and less frequent app categories, the subset ensures that evaluation results remain reflective of real-world deployment conditions. Consequently, this sampled dataset serves as a compact yet reliable benchmark for downstream agent performance analysis.

### E.4 HUMAN EVALUATION STUDY ON CONSISTENCY

To assess the consistency between our synthesized instructions and real user behavior, we conducted an additional human evaluation study. The experimental setup is as follows: we invited four professional colleagues who frequently use music, travel, shopping, and office applications, respectively. Each evaluator was asked to assess 400 instructions sampled from these four domains. For comparison, we also provided them with instruction sets of the same categories from other existing benchmarks. The evaluation criterion was whether an instruction matches what they would naturally say when interacting with their mobile voice assistant. The results are shown in Table 10.

### E.5 DISCUSSION

Table 11: Qwen2-VL, CogAgent, and OS-Atlas evaluated on AITZ-Noise. Metric "Noisy" means in-domain noisy step accuracy. More experiments can be seen in Table 7.

| Agent | Training Data | General | | | Google App | | | Install | | | Web Shopping | | |
|---|---|---|---|---|---|---|---|---|---|---|---|---|---|
| | | Step. Acc | Noisy | SR | Step. Acc | Noisy | SR | Step. Acc | Noisy | SR | Step. Acc | Noisy | SR |
| Supervised Fine-tuning Setting(LoRA) | | | | | | | | | | | | | |
| CogAgent-18B | AITW(CoaT) | 40.4 | - | 11.5 | 38.1 | - | 11.3 | 45.2 | - | 17.3 | 39.1 | - | 13.4 |
| Qwen2-VL-7B | AITZ(CoaT) | 36.1 | - | 8.3 | 39.1 | - | 11.2 | 50.9 | - | 20.7 | 41.8 | - | 15.2 |
| Qwen2-VL-7B | AITZ-Noise | 39.8 | 98.0 | 11.7 | 42.3 | 99.0 | 16.6 | 60.9 | 100 | 30.4 | 41.5 | 99.0 | 13.3 |
| OS-Atlas-7B | AITZ-Noise | 46.2 | 99.0 | 18.7 | 50.2 | 99.5 | 21.3 | 62.4 | 100 | 33.0 | 44.8 | 99.0 | 17.3 |
| Supervised Fine-tuning Setting(Full) | | | | | | | | | | | | | |
| Qwen2-VL-7B | AITZ-Noise | 43.2 | 96.0 | 15.6 | 46.2 | 97.5 | 19.8 | 64.2 | 98.5 | 35.6 | 50.9 | 98.0 | 22.3 |
| OS-Atlas-7B | AITZ-Noise | 47.2 | 98.0 | 19.0 | 47.1 | 99.0 | 22.3 | 66.7 | 99.0 | 38.0 | 51.8 | 99.0 | 23.5 |

**Solving in-domain noise through post-training.** We are more focused on whether increasing the proportion of noisy training data can address the in-domain noisy problem. As shown in Table 11, Qwen2-VL, compared to the original AITZ training data, shows a Step.Acc improvement of 3.7%, 3.2%, 10.0%, -0.3% and SR improvement of 3.4%, 5.4%, 9.7%, -1.9% on the four sub-tasks. At the same time, full parameter fine-tuning outperforms LoRA in overall results but performs slightly worse than LoRA on noise step processing (an average of 1.5% lower). After training, the agent is able to correctly handle the vast majority of noisy steps (with an accuracy greater than 97%), demonstrating the effectiveness of training with noisy data.

**Unusable noises.** We exclude the keyboard-occlusion noise type from our evaluation. Under the current evaluation protocol, VLM agents take both the GUI screenshot and the parsed HTML as input. The handwriting keyboard occludes part of the on-screen content in the GUI image, whereas the same content remains fully accessible in the HTML DOM, leading to inconsistent visibility signals and additional confounding. As shown in Fig. 13, in the dictionary app the words contour and trailer are visually blocked by the handwriting keyboard in the GUI screenshot, but they are still exposed in the parsed HTML.

### E.6 BEHAVIORAL ANALYSIS OF GUI AGENTS.

To better understand the behavioral characteristics of current GUI agents, we conduct an analysis along three major dimensions: repeated loops, hallucinated interactions, and navigation inconsistency. These analyses cover both single-path and multi-path settings and are evaluated across a diverse set of frontier VLM-based agents.

**Analysis of Repeated Loops in Multi-Path Tasks.** In our benchmark, single-path tasks follow a pre-defined, deterministic sequence of actions; therefore, no repeated loops can occur. By contrast, multi-path tasks allow free path selection, making repeated loops a natural indicator of unstable decision-making or inefficient exploration. We adopt an n-gram–based repetition metric to characterize these behaviors. For each trajectory, we compute: (1) **Repeat Length**: average length of repeated action subsequences. (2) **Repeat Count**: average number of repeated subsequences. (3) **Length-2 Count**: frequency of minimal oscillations (e.g., "next–back").

Table 12: Repeated-loop statistics on simple tasks.

| Model | Repeat Length | Repeat Count | Length-2 Count |
| --- | --- | --- | --- |
| Qwen2.5-7B | 2.0845 | 1.6901 | 1.16 |
| Qwen2-72B | 3.0048 | 2.4105 | 1.9657 |
| GPT-4o | 0.6351 | 0.4844 | 0.2375 |
| GLM-4.1 | 1.6714 | 1.2571 | 0.5667 |

Table 13: Repeated-loop statistics on complex tasks.

| Model | Repeat Length | Repeat Count | Length-2 Count |
| --- | --- | --- | --- |
| Qwen2.5-7B | 3.2210 | 2.4842 | 2.00 |
| Qwen2-72B | 4.8108 | 3.4241 | 3.42 |
| GPT-4o | 0.1194 | 0.0895 | 0.04 |
| GLM-4.1 | 1.9142 | 1.2285 | 0.2766 |

**Hallucinated Interactions.** We further study *interaction hallucination*, where an agent predicts the correct action type but executes an incorrect action instance. We quantify hallucination frequency using:

$$\text{Hallucination Ratio} = 1 - \frac{\text{Step Accuracy}}{\text{Type Accuracy}}.$$

A large ratio indicates difficulty grounding predicted actions onto the GUI.

Table 14: Hallucination analysis across simple and complex tasks.

| Model | Cat. | Type S. | Step S. | Halluc. S. | Type C. | Step C. | Halluc. C. |
| --- | --- | --- | --- | --- | --- | --- | --- |
| CogAgent-18B | Single | 75.6 | 20.9 | 72.3 | 62.3 | 20.8 | 66.6 |
| UGround-7B | Single | 73.0 | 39.5 | 45.8 | 73.8 | 36.0 | 51.2 |
| UI-Tars-7B-dpo | Single | 75.3 | 41.8 | 44.4 | 75.9 | 37.8 | 50.1 |
| OS-Atlas-7B-pro | Single | 82.1 | 51.5 | 37.2 | 83.5 | 50.6 | 39.4 |
| Kimi-VL-A3B | Single | 75.1 | 21.7 | 71.1 | 61.8 | 21.5 | 65.2 |
| DeepSeek-VL2 | Single | 72.6 | 38.8 | 46.5 | 73.1 | 35.2 | 51.8 |
| UI-Tars-72B-dpo | Single | 94.3 | 64.2 | 31.9 | 96.0 | 63.5 | 33.8 |
| GUI-OWL-7B | Single | 94.7 | 71.2 | 24.8 | 96.0 | 68.4 | 28.7 |
| UI-TARS-1.5-7B | Single | 98.2 | 72.2 | 26.4 | 97.1 | 77.5 | 20.1 |
| OpenCUA-32B | Single | 98.0 | 73.1 | 25.4 | 97.4 | 76.2 | 21.7 |

**Navigation Inconsistency.** Navigation confusion is evaluated using:(1) **StepAcc / SR**: a high ratio indicates correct micro-actions but poor global navigation. (2) **Path Efficiency**: ratio between actual and optimal path lengths.

We report results on multi-path tasks:

Table 15: Navigation inconsistency analysis on multi-path tasks.

| Model | SE | StepAcc S. | SR S. | StepAcc/SR | StepAcc C. | SR C. |
|---|---|---|---|---|---|---|
| InternVL2-40B | 5.8 | 43.0 | 10.1 | 4.2 | 46.0 | 6.5 |
| Qwen2-VL-72B | 5.2 | 62.8 | 20.6 | 3.0 | 58.9 | 7.0 |
| Qwen-VL-Max | 5.9 | 67.6 | 12.6 | 5.3 | 63.1 | 9.6 |
| GPT-4v | 6.1 | 29.7 | 3.0 | 9.9 | 29.4 | 1.5 |
| GPT-4o | 5.3 | 61.8 | 19.8 | 3.1 | 61.7 | 16.5 |

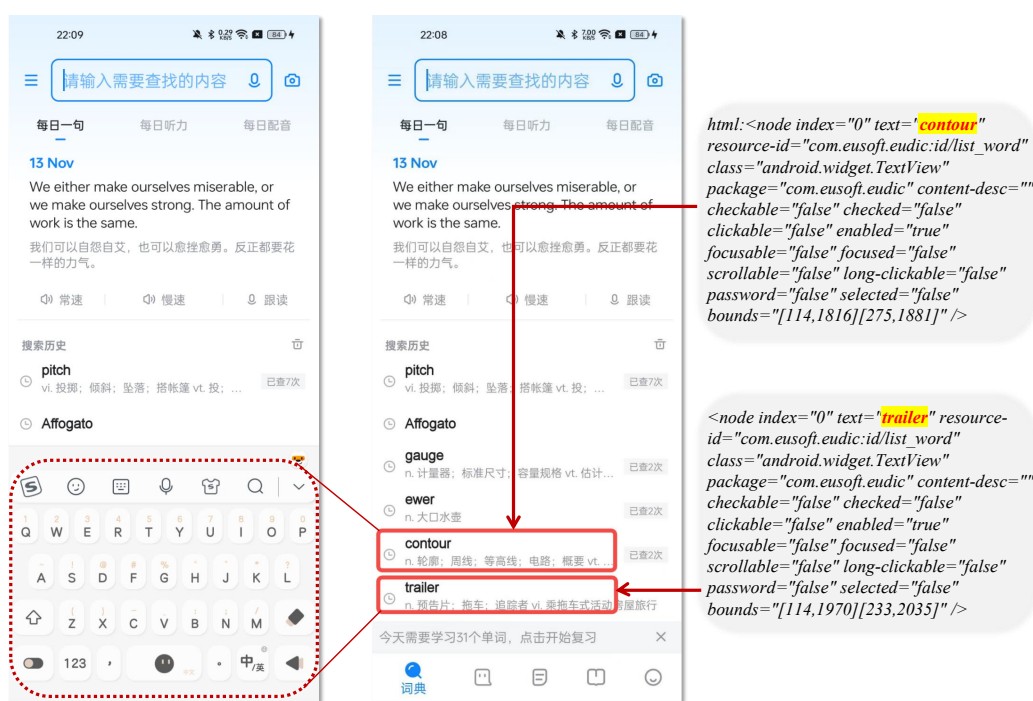

Figure 13: A type of noise rendered unusable when the text and image information conflict.

## E.7 TEST CASE STUDY

**1. Common-Simple**

**2. Common-Complex**

Figure 15 shows a Common-Complex case of SMAN-Bench and the GIAS results are as follows:

1. On the "My Gold" page of the mobile app, click the "Category" tab to enter the book category page.
2. On the category page, select the "Plot" category under the "Boys" tab.
3. In the plot category, select the "Return of the Strong" category to enter the list of books in this category.
4. In the "Return of the Strong" category, select the book "The First War God of the North."
5. On the book details page of "The First War God of the North," click the rating of 8.1 to view the ratings and reviews.
6. On the review page, click "Must-see masterpiece" to view specific book review details.
7. Enter the comment "Science Fiction" on the book review details page and submit it.

Task: Help me find and evaluate a book called *"The First War God of the North"*, view its ratings and related reviews, and add your own feedback under specific reviews.

**Instruction: Help me find the current popular audiobook content and browse different audiobook categories.**

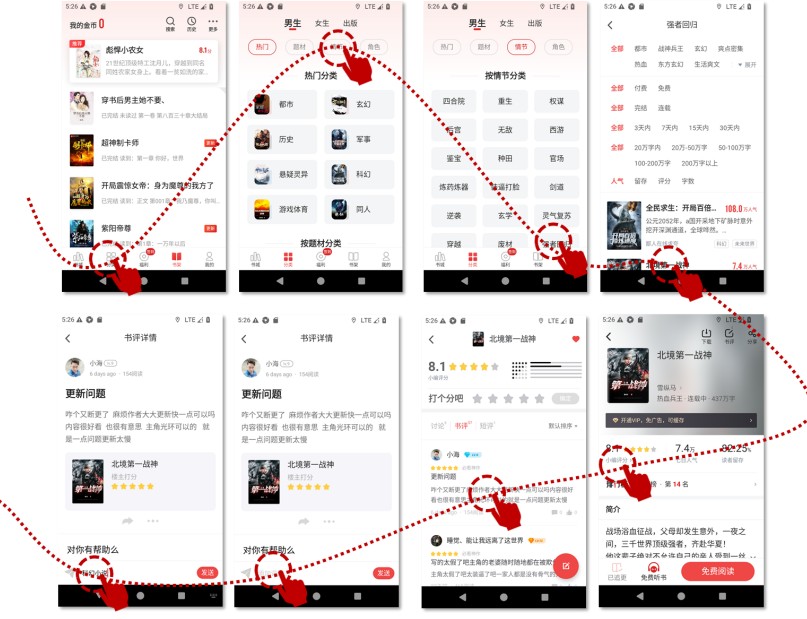

**Step-by-step description:\n1. In the main interface of the music app, click the "Audiobook" option to enter the audiobook area.\n2. In the audiobook page, swipe up to browse different audiobook content.\n3. Continue swiping in the audiobook page to view more audiobook categories and content.\n**

Figure 14: Common-Simple test case.

**Instruction: Help me find and evaluate a book called "The First War God of the North", view its ratings and related reviews, and add your own feedback under specific reviews.**

Figure 15: Common-Complex test case.

**3. Noisy Data**

**4. AITZ-Noisy**

**5. Ambiguous Data**

**Instruction: Please help me find Beijing's air quality index for tomorrow on Caiyun Weather.**

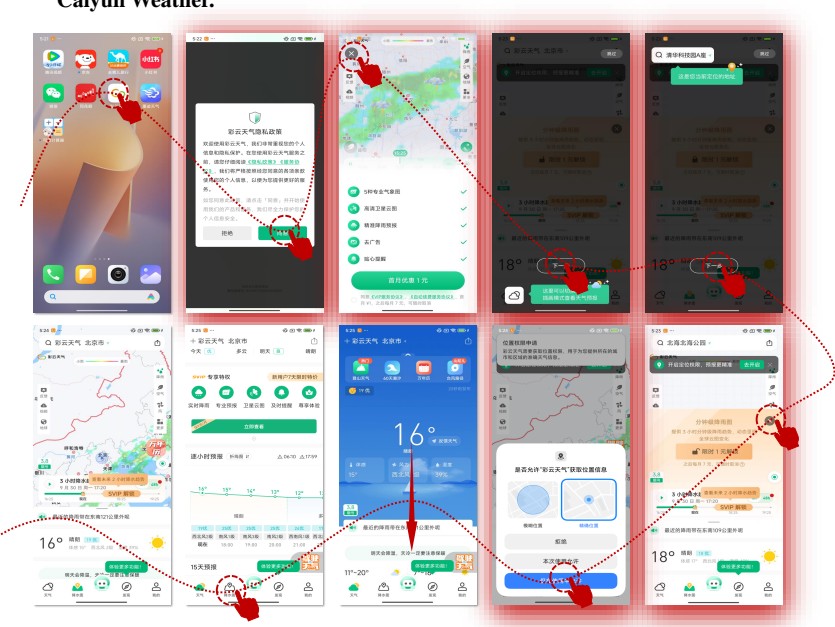

Figure 16: Noisy Data test case. The red shadow in the GUI screenshot are advertisements, pop-ups, or tutorial noise steps.

**Instruction: Search for flights from Mexico city to Boston**

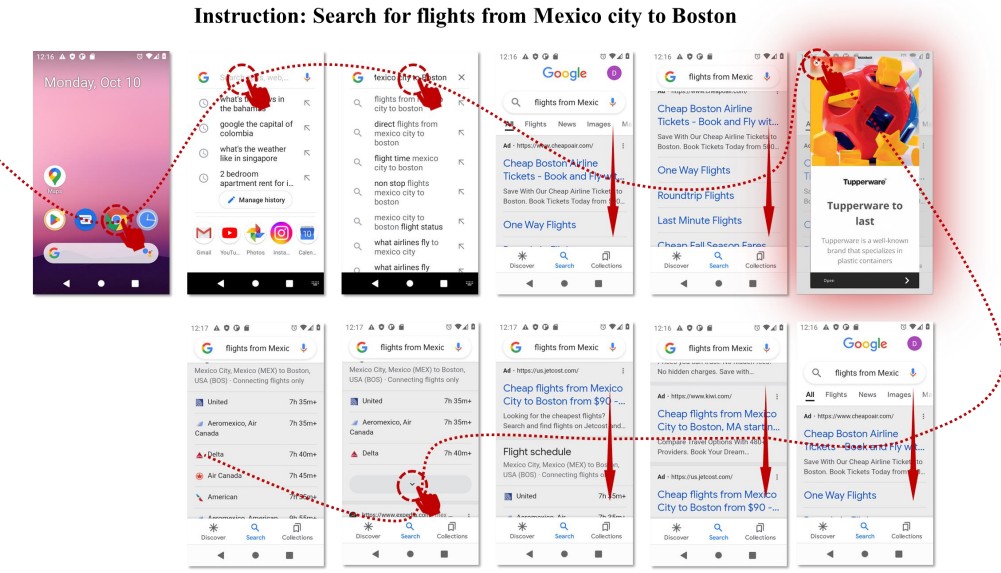

Figure 17: AITZ-Noisy test case. The red shadow GUI screenshot in the trajectory is artificially inserted noise.

**6. Slot-Template Based Instructions** From Table 17, we can observe that the issue of slot-based instructions sounding "programmatic" or "mechanical" indeed exists, and is predominantly concentrated in complex tasks with a larger number of slots. Long and logically structured instructions are difficult to generate through templates, as they typically require coherent dependencies between earlier and later parts of the instruction.

Table 16: Test cases study on SMAN-Bench-Ambiguous.

| Ambiguous Instruction | Find information about a movie |
|---|---|
| Q: Which app should be used? | A: Use Douban. |
| Q: Which app does this page belong to? | A: Douban. |
| Q: In which section should I search? | A: Movies. |
| Q: Do you want to browse a specific ranking? | A: Yes. |
| Q: For which time period? | A: Upcoming releases. |
| Q: How should it be ranked? | A: By popularity. |
| Q: Which movie do you want to check? | A: The most popular upcoming movie. |
| Q: What information do you need? | A: A complete summary. |
| **Full Instruction** | Find the most popular upcoming movie on Douban |
| **Ambiguous Instruction** | Find a midnight snack |
| Q: Which app should be used? | A: Use Ele.me. |
| Q: Would you like to filter by specific snack categories, speed, | A: Find new items from the nearest store that can |
| Q: Any other conditions? | A: deliver within 30 minutes. |
| Q: Do you have a specific price range? | A: No specific price range. |
| Q: Do you have a preferred cuisine or taste? | A: No preference, just quick delivery within 30 minutes. |
| Q: Which section should you search? | A: Food delivery. |
| Q: What are the speed requirements? | A: Within 30 minutes. |
| Q: What are the distance requirements? | A: Nearest store. |
| **Full Instruction** | Find a new delivery item from the nearest store on Ele.me that can deliver within 30 minutes. |

| Type | Slot # | Slot-template based | LLM-vised | Human-annotated | Quality |
|---|---|---|---|---|---|
| Easy case | 1 | Open "QQ Browser" for me. | Open the "QQ Browser" mobile app for me. | Open the "QQ Browser" mobile app for me. | Good |
| Normal case | 2 | Set an alarm for "tomorrow morning" at "7:30". | Use the clock app to set an alarm for "tomorrow morning" at "7:30". | Please wake me up at "7:30 tomorrow morning" using an alarm, and keep the phone in sleep mode before that. | Fair |
| Difficult case | 4 | Book a flight containing "this week", "Friday", and "Beijing" to "Guangzhou". | Book a direct flight from Beijing to Guangzhou on "Friday this week". | Book a direct flight from Beijing to Guangzhou this Friday, prioritizing shortest flight time and secondarily optimal price. | Poor |

Table 17: Instruction complexity across slot count, generation method, and quality level.

**7. Input words** When collecting data, each app category is associated with a predefined query pool containing 200 queries. These predefined inputs are aligned with the category and actual functionality of the corresponding app. Table 18 presents two illustrative (and slightly reduced) examples from these query pools.

## F  LIMITATIONS

Although multi-path validation similar to that on online machines was achieved on SMAN-Bench, the diverse range of text inputs cannot be exhaustively covered, which differentiates it from online machines. Advanced agents such as AutoGLM (Liu et al., 2024) and others deployed by smartphone manufacturers could not be tested due to permission restrictions.

| App Name | Category | Pre-defined Words |
|---|---|---|
| **Ctrip** | Travel | [[Beijing to Shanghai, Chengdu round trip Seoul], [Harbin, Beijing, Japan, New York, Chengdu], [Singapore 4-day tour, Tibet self-driving tour], [Direct flights, Connecting flights], [High-end hotel chains, Budget hotels, Huazhu Club, Marriott, Sheraton], [Popular attractions, Folk culture, Itinerary planning, Food sharing], [Ancient town folk culture, Local festival events], [World heritage sites, National 5A scenic spots], [Ski resorts, Island vacation spots], [Food districts, Night market exploration], [Hiking trails, Mountain camping sites], [Family-friendly routes, Elder-friendly itineraries], [Budget travel tips, Premium customized travel], [Free-travel pitfalls guide, Public transport guide], [New Year travel plans, Weekend short trips], [Local transfer schemes, Best sightseeing order], [Must-eat lists, Street food exploration], [Michelin restaurant recommendations, Seafood buffet picks], [Local breakfast guide, Night-market snacks collection], [Café map, Dessert shop check-ins], [Specialty food rankings, Niche food sharing]] |
| **QQ Music** | Music | [[Jay Chou songs, JJ Lin albums, G.E.M. hit singles], [Faye Wong classics, Beyond classics, Mayday live versions], [New releases, Top singles chart, Pop hits chart], [KTV hot songs, Bar singer selections], [Dance tracks, DJ mixes, Electronic beats], [Wedding BGMs, Love confession songs, Romantic BGMs], [High-quality lossless, Spatial audio, Master tape quality], [Live versions, Remixes, Demo versions], [Instrumental versions, Pure music, Humming tracks], [Copyright versions, Exclusive tracks], [Song ID by listening, Melody-humming recognition], [Lyrics search, Keyword-based song search], [Beat-based search, Mood-based search], [Cover-image search, Audio-snippet search], [Official MVs, Live performances, HD stage videos], [Music short videos, Trending edit BGMs], [Physical albums, Vinyl records, Digital album purchase], [Viral challenge BGMs, TikTok hits, Xiaohongshu trending music], [Gaming battle themes, eSports BGMs], [Study-focus white noise, ASMR white noise], [Children's songs, Early education music], [Yoga/meditation BGMs, Relaxing natural sound effects]] |

Table 18: Pre-defined word groups for Ctrip and QQ Music in English.

# G   USAGE OF LLM STATEMENT

This paper utilized an LLM to improve the clarity and fluency of the text.

