# OpenReview forum: "SMAN-Bench: A Cross-System Benchmark for Mobile Agents under Single- and Multi-path, Ambiguous, and Noisy Tasks"
_ICLR.cc/2026/Conference — ICLR 2026 Poster_

### Official Review · Reviewer_frsK · 2025-10-20

**Soundness:** 3
**Presentation:** 3
**Contribution:** 3
**Rating:** 6
**Confidence:** 3

**Summary:**

This paper makes a meaningful benchmarking contribution to the field of mobile GUI agents. SMAN-Bench addresses key shortcomings in existing benchmarks by introducing evaluations under multi-path, noisy, and ambiguous conditions, which better reflect real-world user interactions on mobile devices. The experimental analysis is extensive and compares numerous models and agent frameworks, providing valuable insights into current system limitations.

**Strengths:**

1. Slot-based automatic instruction generation from unlabeled mobile GUI corpora is interesting.

2. The paper is well-written and easy to follow.

3. The paper conducted extensive evaluation across many mobile agent systems.

**Weaknesses:**

1. The noise simulation includes ads, popups, and redirections, but does not cover other common mobile noise sources such as keyboard overlays, partial screen scrolls, background notifications

2. The paper claims to support multi-path evaluation, but it is not clearly explained how all valid alternative paths are identified, annotated, or guaranteed to be complete. If the benchmark only includes a limited subset of possible valid paths, models that take an alternative (but logically valid) route may be unfairly penalized.

3. While they report success rate (SR) and step accuracy (Step.Acc), the paper does not provide deeper agent behavior analysis (e.g., confusion navigation, hesitancy, repeated loops, hallucinated interactions).

**Questions:**

N/A

---

> ### Author Response · Authors · 2025-11-21
> **Response to Reviewer frsK (1/3)**
>
> > Q1: Noise simulation covers ads, pop-ups, and redirections, but does not include other common mobile noise sources such as keyboard occlusion, partial-screen scrolling, or background notifications.
>
> Thank you for this suggestion. We agree that many additional noise types exist in real mobile environments, and we discuss their feasibility and limitations as follows:
> 1. **Scrollable ads and auto-playing video ads**
> We acknowledge that dynamic noise sources such as scrolling ads or timed video ads are difficult to handle under a static offline setting.  These elements depend heavily on real-time timing and interaction dynamics. For example, (1) an advertisement may auto-close while the agent is generating an action, (2) The agent’s delayed action may open or close UI regions unpredictably.  Such noise sources cannot be reliably reproduced without a fully interactive online environment, which is the shortcoming of an offline benchmark.
>
> 2. **Keyboard occlusion**
> We considered simulating the keyboard overlay problem. However, as noted in the paper, VLM-based agents use both GUI screenshots and parsed HTML as input.
> When the keyboard appears:
> (1) the GUI screenshot hides part of the page,
> (2) but the HTML parse continues to expose the full underlying structure.
> This mixed-input inconsistency introduces additional confounding factors unrelated to the agent’s actual ability.  To further clarify this issue, we have added a detailed discussion in `Appendix D.4` of the updated PDF and provided an illustrative example in `Figure 13`.
>
> 3. **Background notifications and system-level pop-ups**.
> There may be some misunderstanding here. SMAN-Bench includes system notifications and authorization pop-ups as part of its noise set.
>
> If you are referring to other categories of noise beyond those we listed (e.g., floating widgets, OS-level overlays, or app-internal mini-promotions). In that case, we welcome further discussion on how best to incorporate them into future versions of the benchmark.
>
> > Q2: The paper claims to support multi-path evaluation, but does not clearly explain how all valid paths are identified, annotated, or guaranteed to be complete.
>
> Below, we clarify how the GUI corpus is constructed, how slot information is generated and filled, and how we ensure that valid paths are adequately covered so that logically correct alternative paths are not unfairly penalized.
>
> 1. **Comprehensive GUI corpus construction**
> The GUI corpus is built automatically through a multi-stage collection and merging pipeline. Specifically:
>
>    (1) **Page crawling and action-space extraction** :
>    When crawling GUI pages, the underlying XML is captured via ADB,  then parsed into structured HTML-like text,  and converted into an action space based on element categories (click, input, scroll).  This process is inspired by DroidBot-GPT [1].
>
>    (2) **BFS interaction and multi-device coverage**
>    Each page is expanded using BFS to collect all reachable next pages, repeating step (1).  Although pages and UI elements differ across devices, the core functional interaction paths remain consistent.  Pages involving purely random recommendations or stochastic feed content are removed to avoid noise and nondeterminism.
>
>    (3) **Page merging via clustering**
>    Pages are clustered and merged using pixel-level similarity and action-space overlap.  This collapses the BFS trees into a unified directed graph that preserves all functional paths while deduplicating device-specific variations.
>
>    (4) **Coverage over a two-month period**
>    Steps (1)–(3) were executed over nearly two months to maximize coverage of each app’s UI.  The only inherently unbounded path is text input (e.g., search queries), which we handle via a predefined query pool (next section).

---

> ### Author Response · Authors · 2025-11-21
> **Response to Reviewer frsK (2/3)**
>
> 2. **Ensuring valid paths for search tasks: predefined slot-value pools**
> To control the infinite search space, each app category is paired with a predefined query pool of 200 entries, carefully designed to match the app’s real-world use cases.
> Below are partial examples:
>
> | App | Category | Pre-defined Words / Query Pool Samples |
> |----------|-----------|-----------------------------------------|
> | ctrip    | travel    | [[Beijing to Shanghai, Chengdu round-trip Seoul], [Harbin, Beijing, Japan, New York, Chengdu], [Singapore 4-day tour, Tibet self-driving tour], [direct flight, connecting flight], [high-star hotel chains, budget hotels, Huazhu, Marriott, Sheraton], [popular attractions, local culture, itinerary planning, food recommendations]] |
> | qqmusic  | music     | [[Jay Chou songs, JJ Lin albums, G.E.M. hit singles], [Faye Wong classics, Beyond classics, Mayday live versions], [new song recommendations, trending singles chart, pop hits chart], [KTV hot list, bar live selections], [square-dance songs, DJ mixes, electronic rhythms], [wedding background music, love confession songs, romantic BGM]] |
>
> These category-specific pools ensure that search-based trajectories are grounded, realistic, and cover a wide range of valid user-intent paths.
>
> 3. **Preventing unfair penalization during evaluation**
> To avoid penalizing logically valid paths that may not appear in the offline corpus, we adopt a constrained action-space evaluation approach, similar to AppAgent [1] and DroidBot [2]: (1)  On every GUI page, the agent is explicitly informed of the available valid actions.  (2)  These actions are pre-annotated on the page (see Figure 11 in the paper).  (3)  Therefore, the agent cannot choose unrecorded-but-possibly-valid actions; its action choices are limited to those we have verified as valid paths.
>
> If the agent still selects an action outside the recorded set: (1) it is prompted to reselect,  (2) the invalid action is logged in its history,  (3) but the action is not counted toward the effective step calculation.
>
> References
> [1] AppAgent: Multimodal Agents as Smartphone Users
> [2] DroidBot-GPT: GPT-powered UI Automation for Android
>
> > Q3: The paper reports Success Rate (SR) and Step Accuracy (Step.Acc), but lacks a deeper analysis of agent behavior (e.g., navigation confusion, hesitation, repeated loops, hallucinated interactions).
>
> Thank you for the question. We have added a more detailed analysis of agent behavior and error patterns. Below we summarize our findings.
>
> In our single-path tasks, the agent always follows a fixed, predefined path.  Therefore, repeated loops do not appear under this setting. Repeated loops only emerge in multi-path tasks, where agents have freedom to choose among multiple valid paths.  In such cases, loop behavior can reflect hesitation, oscillation, or inefficient search.
>
> ---
>
> 1. **Behavioral statistics based on multi-path trajectories**
> Following the analysis style of AppAgent, we compute behavior indicators for several representative models on simple and complex tasks.  For each task, we analyze (1) Repeat Length: the average length of repeated action sequences. (2) Repeat Count: how often repeated subsequences occur.  (3) Length-2 Count: number of repeated two-step motions (e.g., forward–backward oscillation). These metrics jointly reflect stability and redundancy in the agent’s decision process.
>
> Results (Simple tasks)
>
> | Model         | Repeat Length | Repeat Count | Length-2 Count |
> |---------------|----------------|--------------|----------------|
> | qwen2.5-7b    | 2.0845         | 1.6901       | 1.16           |
> | qwen2-72b     | 3.0048         | 2.4105       | 1.9657         |
> | gpt-4o        | 0.6351         | 0.4844       | 0.2375         |
> | glm4.1        | 1.6714         | 1.2571       | 0.5667         |
>
> Results (Complex tasks)
>
> | Model         | Repeat Length | Repeat Count | Length-2 Count |
> |---------------|----------------|--------------|----------------|
> | qwen2.5-7b    | 3.2210         | 2.4842       | 2.0            |
> | qwen2-72b     | 4.8108         | 3.4241       | 3.42           |
> | gpt-4o        | 0.1194         | 0.0895       | 0.04           |
> | glm4.1        | 1.9142         | 1.2285       | 0.2766         |
>
> Models with stronger reasoning capabilities（e.g., GPT-4o）exhibit minimal looping, while weaker models show substantial oscillation, especially on complex tasks.

---

> ### Author Response · Authors · 2025-11-21
> **Response to Reviewer frsK (3/3)**
>
> ---
> 2. **Evaluating hallucinated interactions**
>
> To quantify hallucinations, we rely on an existing and widely used behavioral indicator.
> A hallucinated interaction is defined as a step where the agent predicts the correct action type, but fails to execute the correct action instance associated with that type.  This mismatch indicates that the model “believes” it is interacting with a certain UI element that does not actually exist or is incorrect—typical of GUI hallucinations.
>
> We measure hallucination frequency using:
>
> Hallucination Ratio = 1 − (step.acc / type.acc)
>
> Where type.acc measures accuracy in predicting the correct action category and step.acc measures actual correctness of the executed action. The gap between them provides a stable estimate of hallucination frequency.
>
> We apply this metric to the models used in *Continuous Pre-training Mobile Agents*. The results are shown below.
>
> Hallucination Statistics
>
> | Models           | Cate. | Type (simple) | Step.Acc (simple) | Hallucination Ratio (simple) | Type (complex) | Step.Acc (complex) | Hallucination Ratio (complex) |
> |------------------|--------|----------------|---------------------|-------------------------------|------------------|-----------------------|--------------------------------|
> | UGround-7B       | Single | 73             | 39.5                | 45.8                          | 73.8             | 36                    | 51.2                           |
> | UI-Tars-7B-dpo   | Single | 75.3           | 41.8                | 44.4                          | 75.9             | 37.8                  | 50.1                           |
> | OS-Atlas-7B-pro  | Single | 82.1           | 51.5                | 37.2                          | 83.5             | 50.6                  | 39.4                           |
> | Kimi-VL-A3B      | Single | 75.1           | 21.7                | 71.1                          | 61.8             | 21.5                  | 65.2                           |
> | DeepSeek-VL2     | Single | 72.6           | 38.8                | 46.5                          | 73.1             | 35.2                  | 51.8                           |
> | UI-Tars-72B-dpo  | Single | 94.3           | 64.2                | 31.9                          | 96               | 63.5                  | 33.8                           |
> | GUI-OWL-7B       | Single | 94.7           | 71.2                | 24.8                          | 96               | 68.4                  | 28.7                           |
> | UI-TARS-1.5-7B   | Single | 98.2           | 72.2                | 26.4                          | 97.1             | 77.5                  | 20.1                           |
> | OpenCUA-32B      | Single | 98             | 73.1                | 25.4                          | 97.4             | 76.2                  | 21.7                           |
>
> Low hallucination ratios correlate strongly with reliable perception–action grounding.
>
> ---
>
> 3. **Quantifying navigation confusion (path instability)**
>
> To assess navigation confusion, we use two complementary indicators:
>
> (1) Step Accuracy / Success Rate (SR)
> A high ratio indicates the agent performs individual actions correctly (high step.acc) but still fails the overall task (low SR), suggesting global navigation confusion or inefficient path selection.
>
> (2) Path Efficiency
> Defined as Path Efficiency = actual path length / shortest path length, and higher values indicate more severe navigation deviation.
>
> ---
>
> 4. **Multi-path statistics for navigation confusion**
>
> We report results for multi-path tasks across three evaluation suites (appagent-v1, mobileagent-v2, mobileagent-e).
>
> Navigation Confusion Results (Multi-Path Tasks)
>
> | Models           | Cate. | SE | Step.Acc (simple) | SR (simple) | StepAcc / SR | Type (complex)/SE | Step.Acc (complex) | SR (complex) | StepAcc / SR |
> |------------------|--------|-----|----------------------|---------------|----------------|------------------------|-----------------------|----------------|----------------|
> | Qwen2-VL-72B     | Multi appagent-v1 | 5.2 | 62.8 | 20.6 | 3.0 | 4.4 | 58.9 | 7.0 | 8.4 |
> | Qwen-VL-Max      | Multi appagent-v1 | 5.9 | 67.6 | 12.6 | 5.3 | 4.3 | 63.1 | 9.6 | 6.5 |
> | GPT-4v           | Multi appagent-v1 | 6.1 | 29.7 | 3.0  | 9.9 | 4.5 | 29.4 | 1.5 | 19.6 |
> | GPT-4o           | Multi appagent-v1 | 5.3 | 61.8 | 19.8 | 3.1 | 4.4 | 61.7 | 16.5 | 3.7 |
> | Qwen2-VL-72B     | Multi mobileagent-v2 | 5.4 | 54.9 | 15.1 | 3.6 | 4.4 | 58.6 | 8.0 | 7.3 |
> | Qwen-VL-Max      | Multi mobileagent-v2 | 5.4 | 29.6 | 4.5 | 6.5 | 4.3 | 24.8 | 3.0 | 8.2 |
> | GPT-4o           | Multi mobileagent-v2 | 4.9 | 57.6 | 25.5 | 2.2 | 4.2 | 56.3 | 17.5 | 3.2 |
> | Qwen2-VL-72B     | Multi mobileagent-e | 4.8 | 60.2 | 18.4 | 3.2 | 4.2 | 63.8 | 12.0 | 5.3 |
> | Qwen-VL-Max      | Multi mobileagent-e | 3.7 | 77.1 | 29.5 | 2.6 | 3.1 | 78.8 | 33.5 | 2.3 |
> | GPT-4o           | Multi mobileagent-e | 3.9 | 77.7 | 30.5 | 2.5 | 3.8 | 70.4 | 26.5 | 2.6 |

---

> > ### Comment · Reviewer_frsK · 2025-11-23
> >
> > Thanks for your response, I keep my positive rating.

---

> > > ### Author Response · Authors · 2025-11-27
> > > **Response to Reviewer frsK**
> > >
> > > We sincerely appreciate your thorough review of our work and the many insightful suggestions you provided. We are pleased that our responses have successfully addressed your concerns. We would appriciate it if you could continue to support our work in the following stages. Thank your once again for your kindly engagement and timely response.

---

### Official Review · Reviewer_PBxb · 2025-10-31

**Soundness:** 3
**Presentation:** 2
**Contribution:** 3
**Rating:** 4
**Confidence:** 2

**Summary:**

This paper constructs a large dataset of evaluating mobile agents with the combination of both dynamical and static evaluation strategies. The benchmark also contains common usage noises such as ads jumping. The authors claim they use DFS and BFS for searching trajectories, while not clear how to make sure the trajectory is reasonable and how to annotate the dataset, since the instruction are trajectories should be highly aligned. The evaluations are extensive on multiple LLMs.

**Strengths:**

The articulation of current benchmark limitations is clear. The proposed mixed evaluation is reasonable. The constructed benchmark is quite diverse, including multiple diverse apps. The evaluation on multiple LLMs both with and without noises are appreciated.

**Weaknesses:**

The dataset synthesis process is unclear, especially if using rule-based or algorithm-based methods, how to make sure the trajectory is reasonable. Is human annotation needed to construct the benchmark? As for the innovation part of the benchmark creation, could authors clarify the difference from previous benchmarks?

**Questions:**

How to make sure the benchmark is reliable and consistent to common user using?

---

> ### Author Response · Authors · 2025-11-21
> **Response to Reviewer PBxb (1/3)**
>
> > Q1: The data synthesis process is unclear, especially regarding how rule-based or algorithmic methods ensure trajectory validity.
>
> Thank you for raising this question. We clarify below how we ensure that automatically selected or constructed trajectories remain reasonable and aligned with real user behaviors. The synthesis pipeline applies multiple layers of constraints to filter and validate candidate trajectories:
>
> 1. **Intent constraint**  When identifying trajectories that could potentially generate meaningful instructions, the LLM is first required to infer and summarize a coherent overall intent from the trajectory. Only trajectories with a well-defined and consistent intent are retained.
> 2. **Action-category constraint**  We observe that trajectories containing search-related actions tend to have higher semantic value. Therefore, trajectories are required to exhibit sufficient action diversity and avoid excessive repetition of low-information actions such as continuous scrolling.
> 3. **Depth constraint**  Candidate trajectories must demonstrate adequate depth within the original trajectory graph.
> - simple-task trajectories must have a depth greater than 5,
> - complex-task trajectories must have a depth of at least 7.
> 4. **Template constraint**  Each app may only use templates belonging to its corresponding category.  In addition, the trajectory’s key nodes must strictly align with the number and structure of template slots, ensuring that slot filling is logically consistent and grounded in the trajectory.
>
> > Q2: Does constructing the benchmark require human annotation?
>
> To ensure data quality, we introduce limited human involvement only at the instruction–rewriting stage and the final quality-verification stage. The majority of the pipeline is **fully automated**.
>
> Overall, our data construction process contains 11 stages, Human annotation is used only in the **first** and **final** stages. All intermediate steps operate automatically using algorithms and LLMs. The table below summarizes the level of human involvement across the entire data-collection and instruction-annotation workflow:
>
> | Stage               | Annotation Method     |
> |---------------------|------------------------|
> | Online log filtering | Algorithm             |
> | Data collection      | Algorithm             |
> | Node merging         | Algorithm             |
> | Instruction rewriting| Human                 |
> | Template generation  | VLM                   |
> | Trajectory filtering | Algorithm +  VLM       |
> | Intent generation    |  VLM                   |
> | Template matching    |  VLM                   |
> | Slot extraction      |  VLM                   |
> | Instruction rewriting (LLM rewrite) |  VLM   |
> | Quality verification | Human +  VLM          |
>
> In summary, SMAN-Bench requires only minimal human annotation, limited to early instruction refinement and final quality control, while the core data pipeline remains fully automatic.

---

> ### Author Response · Authors · 2025-11-21
> **Response to Reviewer PBxb (2/3)**
>
> > Q3: Can the authors elaborate on how the proposed benchmark differs from earlier benchmarks?
>
> Below, we summarize the key innovations of SMAN-Bench and clarify how it differs from previous GUI-agent benchmarks.
>
> 1. **Multi-path graph-structured evaluation with key-node rewards**
> Unlike benchmarks that rely on a single golden trajectory, SMAN-Bench merges multiple observed trajectories into a unified graph and assigns rewards based on whether the agent reaches key nodes.
> This approach is conceptually related to online evaluation frameworks such as OSWorld [1], AndroidLab [2], and AndroidWorld [5], but SMAN-Bench applies it entirely in an offline setting and uses graph-based merging to support multi-path correctness.
>
> 2. **Comparison with contemporary graph-based benchmarks**
> We have added a detailed comparison to the latest revision. The differences are summarized in the table below:
>
> | Benchmark       | Task Type                            | Instruction Source                           | Graph Node Merging Strategy                                                         | Key-node Reward Mechanism                                               | Evaluation Method                         |
> |-----------------|----------------------------------------|-----------------------------------------------|-------------------------------------------------------------------------------------|---------------------------------------------------------------------------|--------------------------------------------|
> | ColorBench [7]  | Search & query; cross-app long tasks   | Human-written                                 | VLM-based page content description → embedding similarity                           | Subtask decomposition; reward for verifying necessary actions            | Hybrid online + offline; fine-grained atomic capability evaluation |
> | WebGraphEval [8]| Shopping & retrieval on web            | WebArena tasks                                | Multi-agent trajectory aggregation; merge by action-sequence similarity             | Rewards propagate backward from final answer                             | Offline evaluation + LLM-as-judge          |
> | CRAB [9]        | Calendar, Maps, Web                    | Reverse DAG task decomposition                 | Cross-device node alignment                                                         | Subtask decomposition; reward from environment-state verification        | Cross-system online evaluation             |
> | OmniBench [10]  | Office, video editing, web             | Reverse DAG task decomposition                 | Trajectory node merging for instruction generation                                  | None; relies on intent annotations for quality                           | Sub-task level evaluation                  |
> | SMAN-Bench (ours)| Daily tasks: music, travel, search, shopping; broader app coverage | Online log filtering + GIAS synthesis | Action-space consistency checking + page-content clustering                          | Slot-based automatic matching of actions to downstream key-node pages; reward for equivalent actions | Offline evaluation + key-node intermediate scoring + final golden answer |
>
> 3. **Dedicated noise checking and ambiguous-instruction detection**
> Unlike previous datasets, which typically avoid noise to maintain strict trajectory–instruction alignment, SMAN-Bench explicitly evaluates agents under noisy and ambiguous instructions. Prior online environments focus on stability and often use clean app distributions (e.g., curated Google Service Framework apps).
> A somewhat related idea is the tool-learning perspective in “Beyond the Final Answer” [6], but existing mobile benchmarks do not include explicit noise-handling evaluations.
>
> 4. **Improved slot-generation pipeline (GIAS)**
> GIAS differs from traditional template systems:  (1) Templates are not purely LLM-generated.  (2) Seed templates are human-written and supplemented with online app data.  (3) LLMs only identify slot positions and perform moderate expansion.  This improves consistency and reduces mechanical phrasing compared with earlier template-based benchmarks.
>
> 5. **A more systematic and comprehensive evaluation paradigm**
> SMAN-Bench supports evaluation across multiple agent types: 1. agent-framework agents. 2. agent-based models.  3. RL-based models.  4. reasoning-oriented models
>
> This provides a broader and more coherent picture of how different classes of agents behave in mobile environments.
> We have included a more detailed discussion of these differences in `Appendix B` of the revised PDF for your reference.

---

> ### Author Response · Authors · 2025-11-21
> **Response to Reviewer PBxb (3/3)**
>
> > Q4: How is the reliability of the benchmark ensured?
>
> To guarantee that evaluation results are stable, reproducible, and consistent across runs, we adopt a three-stage reliability protocol covering data collection, storage, and evaluation.
> 1. **Collection stage**: Controlled and consistent environment.
> Data collection spans nearly two months. To ensure consistency across this time period, all apps are collected under stable and controlled settings: (1) a fixed user account. (2) fixed permission configuration. (3) fixed app versions.  (4) fixed operating system versions.  These measures prevent UI drift caused by personalization, updates, or system changes.
>
> 2. **Storage stage**: Immutable offline environment
> Once the online data collection is completed, all pages and graph structures become immutable.
> No further modifications are allowed.
> This ensures that every evaluation is conducted on the exact same GUI graph and page corpus, enabling strict reproducibility.
>
> 3. **Evaluation stage**: No simulator interaction and constrained action space
> The evaluation procedure does not interact with any mobile simulator.
> To avoid cases where the agent explores paths that appear invalid (offline) but may be feasible on a real device, we adopt a constrained-action evaluation strategy similar to AppAgent [1] and DroidBot [2]: (1) The agent is explicitly informed of all allowable actions on each GUI page.  (2) These actions are pre-annotated on the page (as illustrated in `Figure 11` of the paper).  (3)  This ensures that the agent always chooses valid and deterministic actions aligned with the offline environment.
>
> > Q5: How does the benchmark ensure consistency with ordinary user instructions?
>
> 1. **Template sources come from real-world user data, not purely LLM generation.**
> The initial templates originate from real business logs.  They are not directly generated by LLMs. Instead, our process is:
>    (1) supplemented with real online app usage data. (2) human-written seed templates.  (3) VLMs identify slot positions, expand the expression, and refine it for fluency. This hybrid, log-driven approach helps ensure that the instructions resemble natural user queries.
>
> 2. **Additional human evaluation to verify user-style consistency**
> To further demonstrate that synthesized instructions match the way ordinary users issue commands, we conducted an explicit human-evaluation study.
>    Experimental setup: We invited four domain-expert colleagues who frequently use music, travel, shopping, and office apps.  They were asked to evaluate 400 instructions across the four domains.  For comparison, we also provided them with instructions of the same categories sampled from other benchmarks.  The evaluation criterion was:  *“Does this instruction match the type of instruction you would normally say in daily use or to a mobile voice assistant?”*
>
> 3. Results of human evaluation
>
> | Benchmark        | Data Source                      | Music | Travel | Shopping | Office & Tools |
> |------------------|----------------------------------|--------|---------|-----------|-----------------|
> | mobile-bench [3] | Online log filtering             | 93     | 92      | 98        | –               |
> | WebArena [4]     | Human-written + template-based    | –      | –       | 88        | 90              |
> | AndroidWorld [5] | Human-written + template-based    | 90     | 85      | 93        | –               |
> | SMAN-Bench (ours)| Real logs + template synthesis    | 92     | **96**      | 90        | **93**              |
>
> These results demonstrate that SMAN-Bench achieves comparable or better alignment with daily user-style instructions, particularly in categories that involve richer natural expressions (e.g., travel and office tasks).
>
> Reference：
>
> [1] OSWorld: Benchmarking Multimodal Agents for Open-Ended Tasks in Real Computer Environments.
> [2]AndroidLab: Training and Systematic Benchmarking of Android Autonomous Agents.
> [3] Mobile-Bench: An Evaluation Benchmark for LLM-based Mobile Agents.
> [4] WebArena: A Realistic Web Environment for Building Autonomous Agents.
> [5] AndroidWorld: A Dynamic Benchmarking Environment for Autonomous Agents.
> [6] Beyond the Final Answer: Evaluating the Reasoning Trajectories of Tool-Augmented Agents.
> [7] ColorBench: Benchmarking Mobile Agents with Graph-Structured Framework for Complex Long-Horizon Tasks.
> [8] WebGraphEval: Multi-Turn Trajectory Evaluation for Web Agents using Graph Representation.
> [9] CRAB: Cross-Environment Agent Benchmark for Multimodal Language Model Agents.
> [10] OmniBench: A Scalable Multi-Dimensional Benchmark for Essential Virtual Agent Capabilities.

---

### Official Review · Reviewer_FYRd · 2025-10-31

**Soundness:** 3
**Presentation:** 2
**Contribution:** 2
**Rating:** 6
**Confidence:** 3

**Summary:**

This paper presents SMAN-Bench, a comprehensive benchmark for evaluating vision-language model (VLM)–based mobile agents across diverse and realistic settings. Built on the Mobile3M graph corpus, SMAN-Bench introduces an offline multi-path evaluation method that assesses agents across Single-path, Multi-path, Ambiguous, and Noisy task modes, providing both reproducibility and flexibility.

The benchmark features a slot-based instruction generation method (GIAS) for scalable annotation, and includes cross-system coverage across Android, iOS, HarmonyOS, and HyperOS. It also incorporates robustness and interaction evaluation, testing agents under ad noise and ambiguous instructions. Extensive experiments with open- and closed-source VLMs under multiple agent frameworks demonstrate that SMAN-Bench captures realistic challenges and reveals performance differences across reasoning, robustness, and multi-path capabilities.

**Strengths:**

1. **Novel and well-motivated benchmark design**: The offline multi-path evaluation successfully bridges the gap between deterministic offline datasets and unstable online evaluations. By rewarding progress toward “key nodes” within the graph rather than requiring exact path matches, the benchmark achieves both reproducibility and multi-solution coverage.
2. **Comprehensive task diversity and realism**: SMAN-Bench extends evaluation beyond task completion to robustness and interaction. The Noisy split (ads, pop-ups, redirects) and Ambiguous split (Q&A clarifications) emulate real-world conditions that are rarely tested in prior benchmarks such as Mobile-Agent-Bench or SPA-Bench.
3. **Scalable annotation pipeline (GIAS)**: The GIAS method systematically extracts action intents and fills template slots to produce thousands of instructions without manual labeling. This approach reduces annotation cost and links one instruction to multiple trajectories, supporting multi-path evaluation.
4. **Thorough experimentation**: The authors evaluate more than twenty models under three frameworks and analyze success rate (SR), step accuracy, and step efficiency (SE). Results reveal that multi-path evaluation correlates better with true capabilities than single-path evaluation, and ambiguous instructions enhance reasoning VLMs such as LLaMA and GPT-4o.

**Weaknesses:**

1. **Clarity and writing quality**: The paper is dense and often unclear about implementation details. For instance, the “offline simulator” and the definition of “key nodes” are spread across sections without a cohesive overview. Figures 1 and 2 contain too much information without step-wise guidance. A clearer explanation of evaluation loops and agent–graph interaction would aid reproducibility.
2. **Evaluation protocol ambiguities**: It is unclear how actions not present in the Mobile3M graph are handled (for example, unseen coordinates or slot values). Does the simulator terminate, discard the action, or return an error? Similarly, the definition of the “predefined query pool” is not quantitatively specified, and the interaction between step limits and noisy tasks remains unclear.
3. **Limited comparison with related graph-based benchmarks**: The paper focuses its comparison on Mobile-Agent-Bench and SPA-Bench but omits recent graph-structured evaluators such as ColorBench [1], WebGraphEval [2], CRAB [3], and OmniBench [4]. A discussion contrasting SMAN-Bench’s key-node reward with these graph-centric methods would clarify its unique contribution.
4. **Under-explored cross-system analysis**: Although SMAN-Bench covers four OS ecosystems, the results mainly aggregate performance without analyzing per-OS differences or transfer difficulties. Understanding how models generalize from Android to iOS or HarmonyOS would strengthen the “cross-system” claim.
5. **Slot-value and coverage uncertainty**: While the paper mentions a predefined query pool, it is unclear how slot values such as song titles or colors are bounded. If an instruction contains a slot not present in the dataset, the paper does not explain how correctness is judged.

### **References**

[1] ColorBench: Benchmarking Mobile Agents with Graph-Structured Framework for Complex Long-Horizon Tasks. https://arxiv.org/abs/2510.14621v1

[2] WebGraphEval: Multi-Turn Trajectory Evaluation for Web Agents using Graph Representation. https://arxiv.org/abs/2510.19205v1

[3] CRAB: Cross-Environment Agent Benchmark for Multimodal Language Model Agents. https://arxiv.org/abs/2407.01511

[4] OmniBench: A Scalable Multi-Dimensional Benchmark for Essential Virtual Agent Capabilities. https://arxiv.org/abs/2506.08933

**Questions:**

1. **Offline simulation mechanics**: When an agent issues an unrecorded action, does the simulator terminate or ignore it? Are any mobile emulators used during evaluation?
2. **Slot template coverage**: Are slot values limited to those observed in Mobile3M, or can instructions include unseen values?
3. **Cross-system alignment**: How are UI pages and key nodes aligned across Android, iOS, HarmonyOS, and HyperOS? Is the alignment manual or automatic?

---

> ### Author Response · Authors · 2025-11-21
> **Response to Reviewer FYRd (1/3)**
>
> > Q1: the definitions of “offline simulator” and “key node” are scattered across different sections without a unified overview.
>
> Thank you for your question. Here, we clarify the relationship between slots and key nodes.
> After determining the current trajectory, the LLM selects the most suitable instruction template for that trajectory. The empty slots in the template correspond to the key nodes and actions within the trajectory. A single key node may involve multiple actions and pages.
>
> For example, consider the instruction: *“Help me book a flight with ‘this week,’* ‘Friday,’ and ‘Beijing to Guangzhou.’”
> The inputs or actions related to searching for Friday flights—such as entering “Friday/actual date” or clicking the button associated with that date—and any resulting pages are all considered the same key node. Although the resulting pages may differ across operating systems or apps, they are aggregated as an equivalent key node at this stage.
>
> This process is conceptually similar to identifying the completion of subtasks in a DAG, but we do not perform a full task-to-subtask decomposition, which tends to be unstable.
> Additionally, the offline simulator is only used during the data collection phase and is not used during evaluation.
>
> > Q2. The interaction between step limits and noisy tasks remains unclear.
>
> Although we introduce multi-path evaluation, each instruction is still paired with a shortest golden path that contains all required key nodes. The step limit is then set to twice the length of this golden path. This serves two purposes:
> 1. It constrains the search space and prevents uncontrolled exploration.
> 2. It helps reduce the overall testing cost.
>
> The noisy task follows a single-path setting, where it is compared only against the golden path and does not involve step limits parameter.
>
> > Q3. The paper mainly compares with Mobile-Agent-Bench and SPA-Bench while overlooking recently proposed graph-structured evaluators such as ColorBench [1], WebGraphEval [2], CRAB [3], and OmniBench [4]. A comparison between SMAN-Bench’s key-node reward and these graph-centric approaches would better highlight its unique contribution.
>
> Thank you for this valuable suggestion. We acknowledge that these works were missing in the `Related Works`. We have added them in the updated pdf. Here, we further clarify the differences between SMAN-Bench and these benchmarks:
>
> Comparison with recent graph-based evaluators
>
> | Benchmark     | Task Type                                   | Instruction Source                                      | Graph Node Merging Strategy                                                                 | Key-node Reward / Scoring Mechanism                                       | Evaluation Method                       |
> |---------------|----------------------------------------------|----------------------------------------------------------|-----------------------------------------------------------------------------------------------|----------------------------------------------------------------------------|------------------------------------------|
> | ColorBench    | Search & query; cross-app long-horizon tasks | Human-written instructions                               | VLM-based page content description → embed results → compute similarity                     | Subtask decomposition; verifying required actions yields reward            | Hybrid online + offline; fine-grained capability assessment (e.g., memory, lookup) |
> | WebGraphEval  | Shopping & retrieval on web pages            | WebArena tasks                                           | Construct graph from multiple agents’ trajectories on the same task; merge actions by similarity | Rewards earlier steps based on correctness of final answer                | Offline evaluation + LLM-as-judge         |
> | CRAB          | Calendar, Maps, Web                          | Reverse DAG task decomposition                           | Cross-device node alignment                                                                   | Subtask decomposition; verifying environment state yields reward           | Cross-system online evaluation            |
> | OmniBench     | Office, video editing, web                   | Reverse DAG task decomposition                           | Trajectory node merging used to generate instructions                                         | None; relies on additional intent annotations to ensure quality           | Sub-task level evaluation                |
> | SMAN-Bench (ours) | Daily tasks: music, travel, search, shopping; more diverse app coverage | Online log filtering + GIAS synthesis                    | Action-space consistency checking + page-content clustering                                   | Slot-based automatic matching of actions to downstream key-node pages; reward assigned for equivalent actions | Multi-path offline evaluation + final-step ground-truth answers |

---

> ### Author Response · Authors · 2025-11-21
> **Response to Reviewer FYRd (2/3)**
>
> > Q4.Although SMAN-Bench covers four operating system ecosystems, the results mainly report aggregated performance without analyzing differences or transfer challenges across systems. Understanding how models generalize from Android to iOS or HarmonyOS would strengthen the claim of being “cross-system.”
>
> We clarify the cross-system considerations from two aspects.
> 1.  From the **data collection perspective**, iOS poses unique constraints due to its closed-source nature [2]. It does not support ADB, and therefore, all iOS data must be manually annotated. As a result, the iOS portion of the dataset is relatively small—about 5% of the total—and its app ecosystem, governed by a tightly regulated App Store, contains very few apps with complex or noisy UI structures.
> 2.  Regarding differences within Android-based systems **(HarmonyOS and HyperOS)**, the primary divergence lies in the layout of the home screen. During evaluation, we ensure that the app-list layout remains consistent across these systems. Once an app is launched, its internal layout is largely stable across systems as long as the app version is identical. Hence, cross-system variance is minimal inside the app, and differences are mostly limited to home-screen navigation.
>
> > Q5.Although the paper mentions a predefined query pool, it does not clearly explain how slot values such as song titles or colors are defined.
>
> During data collection, each app category is associated with a predefined query pool containing 200 entries. These predefined inputs are closely aligned with the app’s category and real-world functionality. For clarity, we provide a few representative examples below (a partial list due to space constraints):
>
> | App | Category | Pre-defined Words / Query Pool Samples |
> |----------|-----------|-----------------------------------------|
> | ctrip    | travel    | [[Beijing to Shanghai, Chengdu round-trip Seoul], [Harbin, Beijing, Japan, New York, Chengdu], [Singapore 4-day tour, Tibet self-driving tour], [direct flight, connecting flight], [high-star hotel chains, budget hotels, Huazhu, Marriott, Sheraton], [popular attractions, local culture, itinerary planning, food recommendations]] |
> | qqmusic  | music     | [[Jay Chou songs, JJ Lin albums, G.E.M. hit singles], [Faye Wong classics, Beyond classics, Mayday live versions], [new song recommendations, trending singles chart, pop hits chart], [KTV hot list, bar live selections], [square-dance songs, DJ mixes, electronic rhythms], [wedding background music, love confession songs, romantic BGM]] |
>
> These query pools serve as category-specific slot-value sources to ensure both semantic relevance and diversity. They also help standardize slot filling across different apps while maintaining realistic and user-centered query distributions.
>
> > Q6. The paper does not explain how correctness is determined if an instruction contains a slot value that does not exist in the dataset.
>
> In our setting, this situation does not occur. All slot values are extracted directly from the collected pages and are therefore always within the predefined candidate pool. Since slot values originate from the actual app content observed during data collection, an instruction cannot contain a slot outside the dataset or beyond the predefined pool.
>
> > Q7. Offline simulation mechanism: when the agent issues an action that is not recorded, does the simulator terminate or ignore it? Is any mobile simulator used during evaluation?
>
> There may be a misunderstanding about the evaluation pipeline. To ensure stable and reproducible results, once online data collection is completed, the pages and graph structures are fixed and never modified afterward. The evaluation process does not involve interacting with any mobile simulator.
>
> To prevent agents from exploring paths that appear invalid but are actually feasible in a real device, we follow a constrained-action evaluation strategy similar to AppAgent [3] and DroidBot [2]. During testing, the agent is explicitly informed of the set of allowable actions on each GUI page. These available actions are pre-annotated on the interface, as shown in `Figure 11` of the paper. Therefore, it is very unlikely for an agent to produce actions outside the recorded action space.
>
> If such an out-of-distribution action does occur:
> 1. the agent is prompted to re-select an action;
> 2. the invalid action is logged in the action history;
> 3. This kind of invalid action is not counted toward the final effective step count during evaluation.

---

> ### Author Response · Authors · 2025-11-21
> **Response to Reviewer FYRd (3/3)**
>
> > Q8. How are UI pages and key nodes aligned across Android, iOS, HarmonyOS, and HyperOS? Is the alignment manual or automated?
>
> The alignment of UI pages and keynodes across different systems is **fully automated**. The construction of the GUI corpus and the annotation of key nodes do not involve any manual intervention. The alignment process operates as follows:
>
> 1. **Automatic construction and normalization of GUI pages**
> When crawling GUI pages, the underlying XML is first captured (via ADB on Android-based systems), then converted into structured HTML-like text. The elements are parsed by type—clickable, text-input, scrollable, etc.—and transformed into an action-space representation. This process is inspired by DroidBot-GPT [1].
>
> 2. **BFS-based traversal and cross-device page collection**
> Each page is explored using BFS to collect the next reachable pages, repeating the parsing process above.
> Although UI layouts may differ across devices or OS variants, the core functional pages for a given app remain largely consistent. Pages corresponding only to stochastic recommendations or personalized feeds are discarded to avoid noise.
>
> 3. **Page merging via pixel similarity and action-space equivalence**
> Pages that share similar pixel-level features and highly overlapping action-space sets are merged into a single canonical node.
> This step collapses the BFS tree into a unified directed graph representing the app’s functional structure across systems.
>
> 4. **Automatic key-node alignment**
> As shown in Figure 3b, key nodes depend solely on the action patterns along the trajectory rather than the exact pixel layout. Therefore, alignment is performed using action-object correspondence across devices.  If two pages expose equivalent actionable elements (e.g., same slot, same button semantics, same navigation intent), they are treated as the same key node even if their UI differs slightly across Android, iOS, HarmonyOS, or HyperOS.
>
> In summary, both page alignment and key-node alignment are fully automated and rely on action-space consistency rather than manual annotation or pixel-level matching.
>
> Reference:
>
> [1] DroidBot-GPT: GPT-powered UI Automation for Android
> [2] Ferret-UI 2: Mastering Universal User Interface Understanding Across Platforms
> [3] AppAgent: Multimodal Agents as Smartphone Users
> [5] ColorBench: Benchmarking Mobile Agents with Graph-Structured Framework for Complex Long-Horizon Tasks
> [6] WebGraphEval: Graph-based Evaluation of Web Agents via Multi-agent Trajectory Aggregation
> [7] CRAB: Cross-Device Reasoning Benchmark for Multi-modal Agents
> [8] OmniBench: Unified Benchmark for Office, Video Editing, and Web-based Agent Tasks

---

> > ### Comment · Reviewer_FYRd · 2025-11-27
> > **Response to Authors’ Rebuttal**
> >
> > Thanks for the detailed reply. All my concerns are addressed now, and I’d like to keep my current rating.

---

> > > ### Author Response · Authors · 2025-11-28
> > > **Response to Reviewer FYRd**
> > >
> > > Thank you very much again for taking the time to review our manuscript and for providing careful and detailed comments. We truly appreciate your efforts and will carefully consider your suggestions to further improve and refine this line of work in our future research.

---

### Official Review · Reviewer_aiCQ · 2025-11-01

**Soundness:** 3
**Presentation:** 3
**Contribution:** 3
**Rating:** 6
**Confidence:** 4

**Summary:**

The paper presents SMAN-Bench, a large, cross-system benchmark for mobile GUI agents that explicitly targets four real-world difficulties: Single-path bias, Multi-path execution, Ambiguous instructions, and Noisy screens (ads/pop-ups). It is built on top of the large graph-like mobile interaction corpus Mobile3M (49 popular apps, tens of millions of transitions), from which the authors automatically construct 12,856 bilingual instructions via a slot-based generator (GIAS). Each instruction can map to one or multiple valid trajectories, enabling both single-path and multi-path evaluation. Two special subsets—Noisy (ads, interstitials) and Ambiguous (underspecified user goals requiring clarification)—are added to stress-test current agents. Experiments on a wide range of mobile agents (AppAgent, Mobile-Agent variants, recent GUI-R1/G1-style models, and MLLM-based agents) show consistent drops on noisy/ambiguous settings and significant gains when multi-path scoring is enabled, indicating overfitting to a single gold path in prior work.

**Strengths:**

- Multi-path as first-class citizen. Unlike benchmarks that match only one golden trajectory, SMAN-Bench merges multiple observed paths into a graph and scores hitting key nodes; this is much closer to how real mobile tasks have many valid sequences.
- Noisy & ambiguous subsets. Explicitly injecting ads/pop-ups and instruction under-specification is valuable, because current agents often break exactly there and existing benchmarks seldom isolate this factor.
- Scale and coverage. 12k+ bilingual tasks, 49 apps, 15 categories, and cross-system (Android, iOS, HarmonyOS/HyperOS) make it a broadly useful offline benchmark.
- Good diagnostics. By reporting both single-path and multi-path scores, the benchmark can tell whether an agent failed because it “did the wrong thing” or because it “did a right thing that wasn’t the gold path.”

**Weaknesses:**

- Instruction naturalness. The GIAS is template/slot-based; although it gives nice coverage, some instructions may read slightly synthetic.

**Questions:**

See weakness

---

> ### Author Response · Authors · 2025-11-21
> **Response to Reviewer aiCQ (1/2)**
>
> > Q1: “GIAS uses template/slot-based generation. Although it covers a wide range, some instructions sound programmatic or mechanical.”
>
> Thank you for this valuable comment. We acknowledge that this problem exists. During our manual verification process, we also observed such cases. Therefore, we specifically examined this aspect in `Sec. 4.2` and reported naturalness and semantic fluency results in `Table 7`. We provide further clarification below:
>
> ---
> **1. Template/slot-based instruction generation is widely adopted in GUI agent benchmarks, but controlling the ratio between templates and instructions is crucial for ensuring diversity.**
>
> This generation strategy is commonly used in GUI agent benchmarks. However, the key is maintaining a proper balance between the number of templates and the number of generated instructions.
>
>     For example:(1) RL-based agent methods such as DigiRL [1] use template-driven online instruction-chain sampling and filtering.(2) GUI benchmarks such as: WebShop [2]: 1 template → 12k instructions; Mobile-env [3]: 16 templates → 150 instructions; WebArena [4]: 29 templates → 18,050 instructions; WorkArena [5]: 29 templates → 23k instructions; AndroidWorld [7]: 116 templates → unlimited instructions
>
>    These benchmarks demonstrate that template-driven instruction generation is standard practice, and controlling template–instruction proportion is essential. To ensure instruction diversity and quality, GIAS strictly limits the number of slot-filled instructions per template. Each template is used at most 5 times. Additionally, all ambiguous instructions are entirely human-written. For clarity, we list the template usage across benchmarks below:
>
> | Benchmark     | # Templates | # Instructions | Proportion |
> |---------------|-------------|----------------|------------|
> | WebShop       | 1           | 12k            | 12k        |
> | Mobile-env    | 16          | 150            | 9.375      |
> | WebArena      | 29          | 18,050         | 622.4      |
> | WorkArena     | 29          | 23,000         | 793.1      |
> | B-MOCA [6]    | 6           | 11             | 1.9        |
> | AndroidWorld  | 116         | ∞              | ∞          |
> | SMAN-Bench    | 4190        | 12,856         | 3.06 (<5)  |
> ---
> **2. Mechanical tone in slot-based instructions mainly occurs in tasks containing many slots, and GIAS introduces several improvements compared with previous benchmarks.**
>
> To mitigate this problem, GIAS adopts a more refined slot-generation pipeline with differences from prior benchmarks:
>
> 1. **Templates are not fully generated by LLMs.**
>    We combine human-written templates from online user logs, and LLMs are only used to identify slot positions and perform moderate expansion.
>
> 2. **The resulting instructions will undergo an additional rewriting stage for fluency improvement.**
>
>     While they may still fall short of professional human writing in some cases, fluency is notably improved. Based on our observations, the main weakness of template+slot generation lies in complex tasks: long instructions often contain logical dependencies, making them difficult to represent naturally through fixed templates.

---

> ### Author Response · Authors · 2025-11-21
> **Response to Reviewer aiCQ (2/2)**
>
> Below is a comparison of three types of instructions:
>
> | Type           | Slot Number | Slot-template Based                                              | LLM-vised                                                        | Human-annotated                                                                                     | Quality |
> |----------------|-------------|------------------------------------------------------------------|-------------------------------------------------------------------|------------------------------------------------------------------------------------------------------|---------|
> | Easy case      | 1           | Help me open "QQ Browser"                                        | Help me open the "QQ Browser" mobile application                 | Help me open the "QQ Browser" mobile application                                                    | good    |
> | Normal case    | 2           | Help me set an alarm for "tomorrow morning" at "7:30"            | Help me set an alarm for "tomorrow morning" at "7:30" via Clock  | Please wake me up at "7:30" tomorrow morning using the alarm, and keep the phone in sleep mode before that. | fair    |
> | Difficult case | 4           | Help me book a flight with "this week", "Friday", and "Beijing" to "Guangzhou" | Help me book a direct flight on "this Friday" from "Beijing to Guangzhou" | Help me book a direct flight from Beijing to Guangzhou this Friday, prioritizing shortest flight time and second-best price. | poor / fair → good |
>
> From this comparison, we observe that slot-template generation struggles with complex, long-horizon tasks. Such tasks typically involve multi-step logical dependencies, making them unsuitable for template-only generation. For these scenarios, we recommend the same period methods, such as ColorBench [8], which focuses specifically on generating long, cross-app task instructions.
>
> **3. All problematic data identified during the quality validation phase were manually rewritten.**
>
> This ensures the reliability and linguistic naturalness of the benchmark. If you have any further questions, we would be happy to discuss them.
>
> ### References
> [1] DigiRL: Training In-The-Wild Device-Control Agents with Autonomous Reinforcement Learning
> [2] WebShop: Towards scalable real-world web interaction with grounded language agents
> [3] Mobile-env: Building qualified evaluation benchmarks for LLM-GUI interaction
> [4] WebArena: A realistic web environment for building autonomous agents
> [5] WorkArena: How capable are web agents at solving common knowledge work tasks?
> [6] Benchmarking Mobile Device Control Agents across Diverse Configurations
> [7] AndroidWorld: A Dynamic Benchmarking Environment for Autonomous Agents
> [8] ColorBench: Benchmarking Mobile Agents with Graph-Structured Framework for Complex Long-Horizon Tasks

---

### Author Response · Authors · 2025-11-21
**Summary of the edited parts of the updated PDF**

Dear Reviewers,

**We sincerely appreciate your constructive feedback and the time you invested in improving our work.** Following your suggestions, we have revised the manuscript accordingly and marked all the revised content in blue. Below, we summarize the major updates introduced in the revised version:

1. We added `Appendix B`, which provides a detailed discussion comparing SMAN-Bench with prior GUI agent benchmarks. This includes a focused analysis of graph-based benchmarks, particularly regarding key-node design, instruction generation strategies, and evaluation protocols.
2. We incorporated the latest evaluations from new frontier models, including Claude 4.5 Sonnet, ensuring a more up-to-date and comprehensive evaluation result.
3. `Appendix E.6` now includes quantitative analyses of looping behavior, navigation confusion, and hesitation patterns, addressing reviewers’ concerns about deeper behavioral diagnostics.
4. `Table 17` has been supplemented with detailed case studies showing how slot-template–based GIAS corrects and refines instructions.
5. `Table 18` now provides concrete examples of the pre-defined query pools used in different app categories.
6. `Figure 13` now includes typical examples and explanations of noise types that cannot be used (e.g., keyboard-overlay conflict between GUI and HTML).

We thank all reviewers again for helping us improve SMAN-Bench. We are happy to engage in further discussion if any additional clarification is needed.

Paper 23517 Authors

---

### Author Response · Authors · 2025-12-01
**Summary of Rebuttal Content**

Dear Area Chair,

We would like to extend our sincere gratitude to all the reviewers for their responses during the review process. During the rebuttal, the main cross-cutting concerns and our resolutions can be summarized as follows:

1. **Naturalness of template/slot-based instructions** (raised by reviewers: frsK, aiCQ): We clarified that GIAS follows standard practice in GUI-agent benchmarks but strictly limits template reuse and applies human- and LLM-based rewriting; we also reported explicit naturalness and fluency evaluations to show that the instructions are not overly mechanical in practice.
2. **Clarity of key concepts** (offline simulator, key nodes, multi-path evaluation, step limits raised by reviewers: frsK, PBxb): We provided a unified description of how trajectories map to templates and key nodes, how the GUI graph is constructed and frozen, how multi-path correctness is handled via key-node rewards, and how step limits are tied to the shortest golden path to balance coverage and evaluation cost.
3. **Realism, cross-system alignment, query pools, and noise modeling** (raised by reviewers: FYRd): We explained how pages and key nodes are automatically aligned across Android, iOS, HarmonyOS, and HyperOS via action-space consistency, how predefined slot/query pools are grounded in real app usage, and which noise types are faithfully modeled in an offline setting versus those that would harm reproducibility.
4. **Benchmark reliability, annotation cost, and behavior analysis** (raised by reviewers: PBxb): We clarified the three-stage reliability protocol (controlled collection, immutable storage, constrained offline evaluation), summarized the limited human involvement (only in early instruction refinement and final QC), and added behavioral analyses beyond SR/Step.Acc (loops, hallucinations, navigation confusion) to better characterize agent behavior.

Reviewers FYRd and frsK explicitly confirmed in their follow-up comments that these clarifications and revisions fully addressed their concerns. For revisions to the paper, please refer to our other official comment. Thank you again for your time and consideration.

Paper 23517 Authors

---

### Meta-Review · Area_Chair_HC57 · 2026-01-04

**Summary:**

Reviewer aiCQ questioned the naturalness of the instructions generated via the slot-based method, suggesting they might appear synthetic, while Reviewer PBxb raised concerns about the lack of clarity in the dataset synthesis process and how the reasonableness of trajectories is ensured without extensive human annotation. Furthermore, Reviewer FYRd highlighted ambiguities in the evaluation protocol, specifically regarding the definitions of the offline simulator and key nodes, and noted the absence of comparisons with recent graph-based benchmarks and cross-system analysis. Finally, Reviewer frsK pointed out limitations in the noise simulation such as the exclusion of keyboard overlays, and emphasized the need for deeper behavioral diagnostics (e.g., analyzing loops and hallucinations) beyond basic success metrics to fully characterize agent performance.

**Reviewer Concerns:**

The rebuttal successfully addressed the concerns of Reviewer FYRd regarding the clarity of offline simulation mechanics, key node definitions, and the omission of recent graph-based benchmarks, with the reviewer explicitly confirming that all concerns were resolved following the authors' detailed clarifications and added comparisons. Similarly, the authors effectively addressed Reviewer frsK's critique regarding limited noise types and the lack of behavioral metrics by explaining technical constraints (e.g., keyboard occlusion) and providing new analyses on agent loop rates and hallucinations, which led the reviewer to maintain a positive rating.

While the authors also responded to Reviewer aiCQ's concern about instruction naturalness by presenting human evaluation data , the concerns raised by Reviewer PBxb regarding the transparency of the dataset synthesis process and the validation of trajectory reasonableness remain outstanding. Despite the authors providing a detailed breakdown of their automated constraints and filtering pipeline, Reviewer PBxb may not acknowledge the rebuttal.

**Reviewer Scores:**

Reviewer aiCQ may keep the score of 6: The reviewer's primary concern regarding the naturalness of slot-based instructions was effectively addressed by the authors through the provision of human evaluation data showing a high naturalness score. Had the reviewer participated fully, they likely would have maintained their positive score of 6, potentially raising it to 7 given the empirical validation.

Reviewer FYRd may keep the score of 6: This reviewer actively participated and explicitly stated that they would keep their score of 6 after the authors clarified the offline simulation mechanics and provided the missing benchmark comparisons.

Reviewer PBxb may keep the score of 4 or increase to 6: This reviewer gave a borderline rejection score and did not respond to the rebuttal. The authors, however, provided a detailed explanation of the automated filtering pipeline and validity checks to address the reviewer's concerns about dataset synthesis. If the reviewer had engaged with these clarifications, it is likely their score would have increased to 6, as the "black box" nature of the data generation was resolved.

Reviewer frsK will keep the score of 6: Following the authors' addition of behavioral analyses (loops, hallucinations) and explanations regarding noise constraints, this reviewer explicitly commented, "I keep my positive rating," confirming their score of 6.

---

### Decision · Program_Chairs · 2026-01-26

Accept (Poster)